# The testis protein ZNF165 is a SMAD3 cofactor that coordinates oncogenic TGFβ signaling in triple-negative breast cancer

Zane A Gibbs[1,2], Luis C Reza[1,2], Chun-Chun Cheng[1,2], Jill M Westcott[3], Kathleen McGlynn[1,2], Angelique W Whitehurst[1,2]*

[1]Department of Pharmacology, University of Texas Southwestern Medical Center, Dallas, United States; [2]Harold C. Simmons Comprehensive Cancer Center, University of Texas Southwestern Medical Center, Dallas, United States; [3]Hamon Center for Therapeutic Oncology Research, University of Texas Southwestern Medical Center, Dallas, United States

**Abstract** Cancer/testis (CT) antigens are proteins whose expression is normally restricted to germ cells yet aberrantly activated in tumors, where their functions remain relatively cryptic. Here we report that ZNF165, a CT antigen frequently expressed in triple-negative breast cancer (TNBC), associates with SMAD3 to modulate transcription of transforming growth factor β (TGFβ)-dependent genes and thereby promote growth and survival of human TNBC cells. In addition, we identify the KRAB zinc finger protein, ZNF446, and its associated tripartite motif protein, TRIM27, as obligate components of the ZNF165-SMAD3 complex that also support tumor cell viability. Importantly, we find that TRIM27 alone is necessary for ZNF165 transcriptional activity and is required for TNBC tumor growth in vivo using an orthotopic xenograft model in immunocompromised mice. Our findings indicate that aberrant expression of a testis-specific transcription factor is sufficient to co-opt somatic transcriptional machinery to drive a pro-tumorigenic gene expression program in TNBC.

*For correspondence:
angelique.whitehurst@
utsouthwestern.edu

Competing interests: The authors declare that no competing interests exist.

## Introduction

Tumors frequently activate genes whose expression is normally restricted to the testes (*Sahin et al., 2000*; *Atanackovic et al., 2007*; *Wang et al., 2016*). Since the testes are immune-privileged, the protein products encoded by these genes can evoke an immune response when expressed in tumors (*van der Bruggen et al., 1991*; *Sahin et al., 1995*; *Fijak and Meinhardt, 2006*). Given this feature, these ~270 genes are collectively termed cancer/testis (CT) genes and their encoded proteins are known as CT antigens (*Simpson et al., 2005*). CT antigens have long been considered attractive targets for immunotherapy in a wide variety of cancers (*Whitehurst, 2014*; *Thomas et al., 2018*). In addition, recent work from multiple groups has found that many CT antigens can support neoplastic behaviors in tumor cells (*Gibbs and Whitehurst, 2018*). Specifically, CT antigens have been implicated in supporting mitotic progression, degradation of tumor suppressors, promoting DNA repair and thwarting apoptosis (*Whitehurst et al., 2010*; *Hosoya et al., 2011*; *Cappell et al., 2012*; *Maxfield et al., 2015*; *Pineda et al., 2015*; *Viphakone et al., 2015*; *Nichols et al., 2018*; *Gallegos et al., 2019*). Collectively, studies on CT antigen function over the last 15 years indicate that these proteins are not merely innocuous by-products of aberrant tumor gene expression programs, but rather integral agents of tumor growth and survival.

Using a large-scale, loss-of-function screening approach, our group previously identified the CT antigen, ZNF165, as essential for triple-negative breast cancer (TNBC) survival (*Maxfield et al., 2015*). Depletion of ZNF165 in vitro or in vivo reduces survival of TNBC cells. Importantly, elevated

ZNF165 mRNA expression correlates significantly with reduced survival time in breast cancer, indicating that expression of this protein may confer an aggressive tumorigenic phenotype (*Maxfield et al., 2015*).

ZNF165 is a $C_2H_2$ zinc finger transcription factor that contains an N-terminal SCAN domain and six canonical $Zn^{2+}$ fingers at its C-terminus. Consistent with its characterization as a CT antigen, ZNF165 expression is restricted to human testes and is extremely low or undetectable in normal breast tissues (*Tirosvoutis et al., 1995*; *Maxfield et al., 2015*). Unfortunately, ZNF165 lacks conservation in mice, thereby precluding genetic studies that could illuminate its normal function within the testes (*Tirosvoutis et al., 1995*). However, investigation of its function in TNBC revealed that ZNF165 associates with chromatin to regulate expression of its target genes, which includes approximately 25% of all TGFβ-responsive genes in TNBC cells (*Maxfield et al., 2015*).

The cellular response to TGFβ is highly context selective, often resulting in dramatically different phenotypes. For example, TGFβ is well known to facilitate growth inhibition in breast epithelia and most other somatic tissues, primarily through transcriptional activation of cyclin-dependent kinase inhibitors that result in cell cycle arrest (*Hannon and Beach, 1994*; *Polyak et al., 1994*; *Morikawa et al., 2016*). Conversely, TGFβ has also been shown to stimulate the growth and differentiation of multiple cell types, including chondrocytes, hematopoietic stem cells, and endothelial cells (*Goumans et al., 2002*; *Li et al., 2005*; *Challen et al., 2010*). These contrasting responses to TGFβ are generally dictated by cell type-specific transcription factors, which associate with TGFβ-responsive SMAD proteins (SMAD3 and SMAD4) and recruit them to appropriate target genes for transcriptional regulation (*Chen et al., 2002*; *Gomis et al., 2006*; *Mullen et al., 2011*; *Massagué, 2012*). In TNBC, TGFβ induces a metastatic phenotype by promoting survival, chemoresistance, and the epithelial-to-mesenchymal transition (*Massagué, 2008*; *Bhola et al., 2013*). These processes are generally attributed to the activity of SMAD3, which binds directly to chromatin in a complex with SMAD4. Notably, SMAD3 expression is elevated in late-stage breast tumors compared to other receptor-activated SMADs (*Tian et al., 2003*; *Padua et al., 2008*; *Petersen et al., 2010*).

Here, we investigate the mechanisms by which ZNF165 modulates TGFβ-dependent transcription in TNBC. We find that ZNF165 co-occupies a subset of SMAD3 genomic binding sites and physically interacts with SMAD3 on chromatin, enhancing SMAD3 recruitment to specific loci. We also identify ZNF446 and TRIM27, two proteins never implicated in TGFβ signaling or TNBC, as critical components of the ZNF165-SMAD3 transcriptional complex. Importantly, loss of function of this complex, through genetic inhibition of TRIM27, attenuates growth of an orthotopic xenograft model of TNBC in vivo. These findings establish that anomalous expression of ZNF165 promotes the assembly of a neomorphic transcriptional complex that contributes to oncogenic TGFβ transcription in TNBC.

## Results

### SMAD3 is enriched at ZNF165 binding sites in TNBC cells

To comprehensively investigate whether ZNF165 interfaces with the TGFβ transcriptional network, we first generated a SMAD3 genomic binding profile in WHIM12 cells, where ZNF165 is essential for growth (*Maxfield et al., 2015*). WHIM12 cells were originally developed from a patient-derived xenograft established from a triple-negative, highly chemo-resistant tumor (*Li et al., 2013*). Moreover, these cells are classified within the claudin-low molecular subtype of breast cancer, which exhibits activated TGFβ signaling (*Prat et al., 2010*; *Asiedu et al., 2011*; *Sabatier et al., 2014*). We found that SMAD3 binding is associated with over >27,000 sites in WHIM12 cells, comparable to previous studies of SMAD3 binding in TNBC cell lines (*Tufegdzic Vidakovic et al., 2015*). Intersecting SMAD3 and ZNF165-associated genes revealed that >90% of ZNF165 target genes are also bound by SMAD3, an overlap representing a statistically significant enrichment (p=5.5e-120, hypergeometric distribution) (*Figure 1A*). This is in agreement with our previous findings that ZNF165 targets a significant fraction of the TGFβ-responsive transcriptome (*Maxfield et al., 2015*). Mapping the distance between binding sites further revealed that 90% of shared targets were bound by both transcription factors at a distance of less than 100 kb from one another (*Figure 1B*). Furthermore, 36% of shared target genes had ZNF165 and SMAD3 binding sites within 1 kb, a proximity at which both factors exhibited overlap on chromatin due to the broad enrichment profile of SMAD3 as observed within our dataset (*Figure 1B–E*). We next established ZNF165 and SMAD3 binding

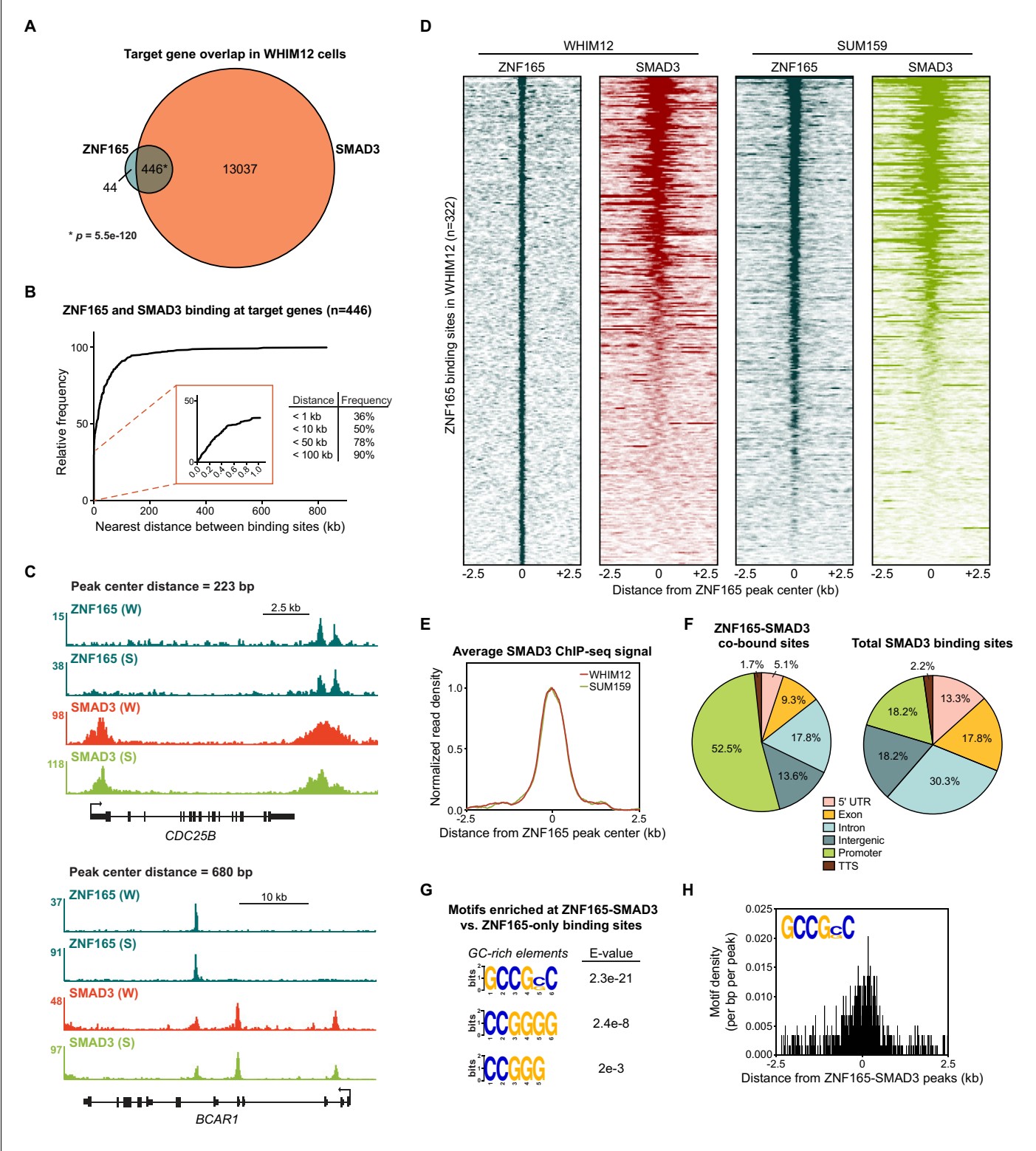

**Figure 1.** SMAD3 is enriched at ZNF165 binding sites in TNBC cells. (**A**) Venn diagram displaying the overlap between ZNF165 and SMAD3 target genes in WHIM12 cells as identified by GREAT (ver 3.0.0). Significance was determined using the hypergeometric distribution. (**B**) Cumulative frequency distribution of the nearest distance between ZNF165 and SMAD3 binding sites associated with each shared target gene. Distance was calculated using the peak coordinates of each factor nearest to one another. (**C**) Browser tracks of ChIP-seq data showing peaks for ZNF165 and SMAD3 near the *CDC25B* and *BCAR1* loci with overlapping distances of 223 and 680 bp, respectively. (**W**) WHIM12, (**S**) SUM159. (**D**) Heatmaps of ChIP-seq data for

*Figure 1 continued on next page*

Figure 1 continued
ZNF165 and SMAD3 in the indicated TNBC cell lines. All peaks within each heatmap are centered ±2.5 kb from the ZNF165 peaks identified in WHIM12 cells (n = 322). (E) Normalized read density (per bp per peak) for SMAD3 plotted ±2.5 kb from the ZNF165-SMAD3 co-bound peaks identified in WHIM12 cells (n = 118). (F) Pie charts displaying the distribution of genomic features bound by ZNF165-SMAD3 (n = 118) or only SMAD3 (n = 27,979) in WHIM12 cells. (G) Motifs enriched at ZNF165-SMAD3 co-bound sites within WHIM12 cells. The 204 ZNF165 binding sites not occupied by SMAD3 were used as a control set of sequences to identify differentially enriched motifs at the shared sites (n = 118). (H) Motif density (per bp per peak) for the GCCG(G|C)C motif plotted ±2.5 kb from the ZNF165-SMAD3 co-bound peaks in WHIM12 cells.

The online version of this article includes the following figure supplement(s) for figure 1:

**Figure supplement 1.** SMAD3 is enriched at ZNF165 binding sites in TNBC cells.

profiles in SUM159 cells, which are also classified within the claudin-low molecular subtype and require ZNF165 for viability (*Prat et al., 2010*; *Maxfield et al., 2015*). SMAD3 enrichment at the ZNF165 binding sites here closely resembled that of WHIM12 cells (*Figure 1C–E*; *Figure 1—figure supplement 1A–C*). Further analysis of SMAD3 binding in MDA-MB-231 cells corroborated these findings (*Figure 1—figure supplement 1D–F*). Together these data demonstrate that despite differences in the genetic backgrounds of TNBC tumor cells, ZNF165 and SMAD3 exhibit significant co-occupancy on chromatin throughout the genome.

We identified 118 sites where ZNF165 and SMAD3 were bound within 1 kb of each other in WHIM12 cells (*Figure 1—figure supplement 1B*, *Supplementary file 1*). More than 50% of these ZNF165-SMAD3 co-bound sites were present in gene promoters (−1 kb or +100 base pairs from TSS) while the majority of SMAD3 binding sites were generally distributed across different genomic regions (*Figure 1F*). Motif enrichment analysis revealed a significant presence of GC-rich motifs at the ZNF165-SMAD3 co-bound regions (n = 118) compared to those bound by ZNF165 alone (n = 204) (*Figure 1G and H*). Such GC-rich SMAD binding elements (SBEs) have recently been characterized as universal binding sites for receptor-activated SMAD proteins, and SMAD3 can bind variants of these motifs with high affinity (*Martin-Malpartida et al., 2017*). Thus, the GC-rich SMAD3 binding element may be an underlying genomic feature of the ZNF165-occupied sites that facilitates co-occupancy. Collectively, these data suggest that ZNF165 and SMAD3 may form cis-regulatory modules on chromatin to direct TGFβ-mediated gene expression in TNBC. In support of this notion, gene set enrichment analysis of ZNF165-responsive genes in WHIM12 cells revealed a significant enrichment of genes involved in the epithelial-to-mesenchymal transition (EMT), a SMAD3-dependent process that promotes metastasis in response to TGFβ signaling (*Figure 1—figure supplement 1G*, *Supplementary file 2*; *Tian et al., 2003*; *Xu et al., 2009*; *Petersen et al., 2010*). Consistent with this, elevated ZNF165 expression in patients with breast cancer is associated with a greater metastatic potential (Hazard Ratio (HR) = 1.4; p=0.004) (*Figure 1—figure supplement 1H*).

## ZNF165 and SMAD3 cooperate to modulate TGFβ-responsive gene expression

Given the enrichment of SMAD3 and ZNF165 at shared sites and their general proximity to one another on chromatin, we next asked whether these transcription factors cooperate to regulate expression of co-bound genes. We leveraged a previously generated gene expression data set of TGFβ responsiveness in WHIM12 cells (*Maxfield et al., 2015*). Here, we assembled a set of 65 TGFβ-responsive genes that are co-bound by SMAD3 and ZNF165 (*Figure 2—figure supplement 1A*). We then compared the impacts of ZNF165 and SMAD3 depletion on a subset of these genes. This analysis revealed a significant correlation (r = 0.52) between expression fold-changes in response to depletion of either transcription factor (*Figure 2A*). In particular, ZNF165 and SMAD3 were required to either activate or repress expression of the majority of shared target genes (quadrant I and III, *Figure 2A*). We also observed opposing effects of SMAD3 and ZNF165 on a smaller subset of genes, suggesting that in some cases, the activities of these two proteins may oppose one another (quadrant II and quadrant IV, *Figure 2A*).

The observed expression changes in *Figure 2A* suggest two modes of regulation for ZNF165-SMAD3 targets. First, ZNF165 collaborates with SMAD3 to activate or oppose TGFβ-mediated gene expression changes (quadrant I and III, *Figure 2A*). Alternatively, ZNF165 can antagonize SMAD3-mediated activation or repression of shared target genes in response to TGFβ (quadrant II and IV,

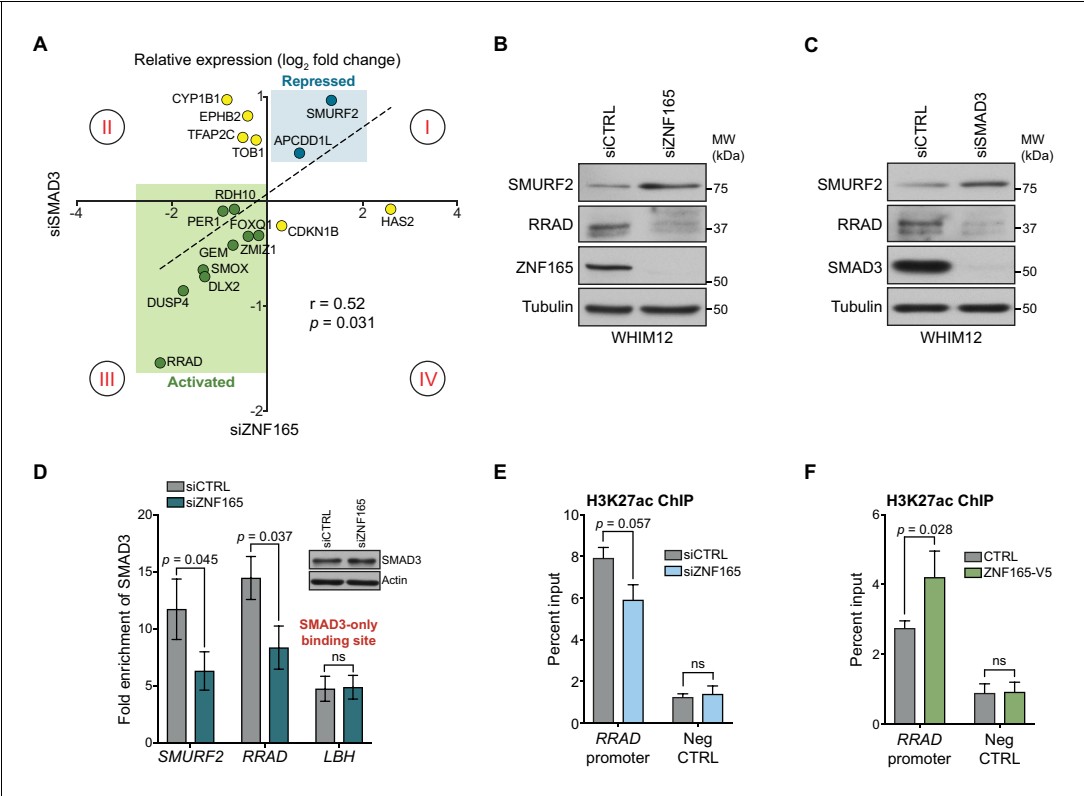

**Figure 2.** ZNF165 and SMAD3 cooperate to modulate TGFβ-responsive gene expression. (A) WHIM12 cells were transfected with siRNA for 48 hr, and qPCR was used to quantify relative expression (log$_2$ fold change) of ZNF165-SMAD3 target genes upon depletion of ZNF165 (x-axis) or SMAD3 (y-axis). The Pearson correlation coefficient is indicated by r. Data are representative of four independent experiments. (B) WHIM12 cells were transfected with siRNA for 72 hr. Whole cell lysates were collected and immunoblotted for the indicated proteins. Data are representative of two independent experiments. (C) As in (B) except cells were transfected with siRNA targeting SMAD3. (D) SUM159 cells were transfected with siRNA for 32 hr and subjected to SMAD3 ChIP followed by qPCR using primers targeting the indicated binding sites. Fold enrichment was determined by dividing the percent input values for SMAD3 by those for IgG. Error bars represent mean ± SEM. P-values were calculated using an unpaired, two-tailed Student's t-test. Data are representative of four independent experiments. (E) SUM159 cells were transfected with siRNA for 32 hr and subjected to H3K27ac ChIP followed by qPCR using primers targeted to the *RRAD* promoter or a negative control region. Error bars represent mean + SEM. P-value was calculated using an unpaired, two-tailed Mann-Whitney test. Data are representative of four independent experiments. (F) SUM159 cells stably expressing ZNF165-V5 or control cDNA were subjected to H3K27ac ChIP followed by qPCR using primers targeted to the *RRAD* promoter or a negative control region. Error bars represent mean + SEM. P-value was calculated using an unpaired, two-tailed Mann-Whitney test. Data are representative of four independent experiments.

The online version of this article includes the following figure supplement(s) for figure 2:

**Figure supplement 1.** ZNF165 and SMAD3 activate RRAD expression to drive neoplastic behaviors in TNBC.

*Figure 2A*). The majority of genes fell into the former category, with *SMURF2* and *RRAD* being the two most responsive and were thus selected as archetypes for follow-up analysis. Importantly, we found that RRAD and SMURF2 protein accumulation corresponded to mRNA alterations upon ZNF165 or SMAD3 depletion (*Figure 2B and C*).

Given the observed collaborative function of ZNF165 and SMAD3 on target gene expression, we next asked whether their association with chromatin was cooperative. ZNF165 knockdown resulted in a ~ 50% reduction of SMAD3 enrichment at the ZNF165 binding sites proximal to both *SMURF2* and *RRAD* (*Figure 2D*). Importantly, SMAD3 binding at the promoter of *LBH*, a site not co-bound by ZNF165, was unaffected by ZNF165 depletion. SMAD3 commonly recruits the histone acetyltransferase, p300, to activate gene expression of its targets (*Feng et al., 1998*; *Janknecht et al., 1998*). Consistent with this, we observed a reduction of H3K27ac at the ZNF165-SMAD3 binding site proximal to *RRAD* in response to ZNF165 knockdown (*Figure 2E*). Conversely, an increase in H3K27ac was detected upon ZNF165 overexpression (*Figure 2F*). These findings suggest that ZNF165 is essential for recruitment of SMAD3 to shared target gene promoters and transcriptional activation.

We reasoned that the coordinated activity of ZNF165 and SMAD3 could modulate expression of genes required for tumorigenic phenotypes. Indeed, SMURF2 is a well characterized negative regulator of the TGFβ pathway whose expression is activated in a TGFβ-dependent, SMAD-independent manner by PI3K signaling (*Ohashi et al., 2005*). Together with our previous findings, data presented here now indicate that ZNF165 coordinates SMAD3 to repress SMURF2 and thereby attenuate negative feedback of TGFβ signaling in TNBC (*Maxfield et al., 2015*). In addition, a report from 2001 indicated that RRAD is sufficient to promote proliferation of breast tumor cells in vitro and in vivo (*Tseng et al., 2001*). Based on this observation, we investigated breast cancer gene expression data sets for clinical correlates with RRAD expression. In the mesenchymal subtype of TNBC, which is enriched for active TGFβ signaling and an EMT gene expression signature, RRAD expression correlated with reduced overall survival (HR = 2.31; p=0.04) and greater metastatic potential (HR = 3.18; p=0.05) (*Figure 2—figure supplement 1B*; *Lehmann et al., 2011*). Conversely, in luminal tumors, which are traditionally ER$^+$ and exhibit tumor-suppressive TGFβ signaling, RRAD expression correlated with improved overall survival (HR = 0.63; p=0.01) and no correlation was observed with distant metastasis free survival (HR = 1.12; p=0.55) (*Figure 2—figure supplement 1C*; *Perou et al., 2000*; *Sato et al., 2014*). Moreover, DepMap analysis revealed that RRAD perturbation generally leads to a reduction in 2D growth of TNBC cell lines, and we found that RRAD was essential for growth of SUM159 cells in soft agar (*Figure 2—figure supplement 1D and E*). However, we observed an enhanced growth phenotype in ER$^+$ MCF7 cells, supporting the notion that TGFβ mediates tumor-suppressive phenotypes in more differentiated breast cancer cells and pro-tumorigenic effects in less differentiated cell types (*Figure 2—figure supplement 1F*; *Tian et al., 2003*; *Massagué, 2008*). Collectively, these data suggest that ZNF165 is capable of directing SMAD3 recruitment to TGFβ-responsive target genes, whose altered expression can support malignant behaviors in the context of TNBC.

## ZNF165 physically associates with SMAD3 in a TGFβ-dependent manner

As ZNF165 appears to facilitate SMAD3 recruitment to chromatin, we asked whether these proteins physically associate. Interaction studies using co-expression/co-immunoprecipitation indicated an association between ZNF165 and SMAD3 (*Figure 3A*). In addition, we found that ZNF165 interacts with the active, phosphorylated form of SMAD3 on chromatin in SUM159 cells stably expressing ZNF165-V5 (*Figure 3B*). Activation of SMAD3 and its resulting nuclear translocation is dependent on the phosphorylation of its C-terminus by TGFβRI in response to TGFβ (*Massagué, 2012*). Thus, we employed the serine/threonine kinase inhibitor SB-431542, which selectively inhibits TGFβRI and downstream phosphorylation of SMAD3, to assess whether the ZNF165-SMAD3 interaction is dependent on pathway activation (*Figure 3C*; *Inman et al., 2002*; *Massagué, 2012*). Proximity ligation assays (PLAs) in SUM159 cells treated with TGFβ revealed an endogenous association between ZNF165 and SMAD3, which was significantly diminished by pre-treatment with SB-431542 (*Figure 3D*). Furthermore, we found that endogenous ZNF165 and SMAD4 also interact in a TGFβ-dependent manner (*Figure 3E*). Importantly, ChIP-seq analysis did not reveal the presence of ZNF165 binding sites near the *SMAD3* or *SMAD4* gene loci (*Maxfield et al., 2015*). This supports the notion that ZNF165 functions to influence SMAD3/4 transcriptional activity via physical association and not through regulation of SMAD3/4 gene expression itself. Taken together, these results indicate that in response to TGFβ, ZNF165 associates with phosphorylated SMAD3 to modulate the transcriptional output of TGFβ signaling in TNBC.

## ZNF446 is an obligate component of the ZNF165-SMAD3 transcriptional complex

We reasoned that ZNF165 and SMAD3 likely function within a multi-protein transcriptional complex to mediate gene expression changes in response to TGFβ. Therefore, we searched for additional ZNF165 interactors using proteomics data generated from a systematic yeast two-hybrid (Y2H) screening approach (*Rolland et al., 2014*). This search returned 48 putative ZNF165 interacting partners, 17 of which are likely transcriptional regulators based on the presence of known gene regulatory domains (*Supplementary file 3*). To triage these candidates, we focused on those containing a SCAN (SRE-ZBP, Ctfin51, AW-1 and Number 18) domain, which is well-documented to mediate specific interactions among SCAN-zinc finger proteins (ZFPs) and the only domain within ZNF165 aside

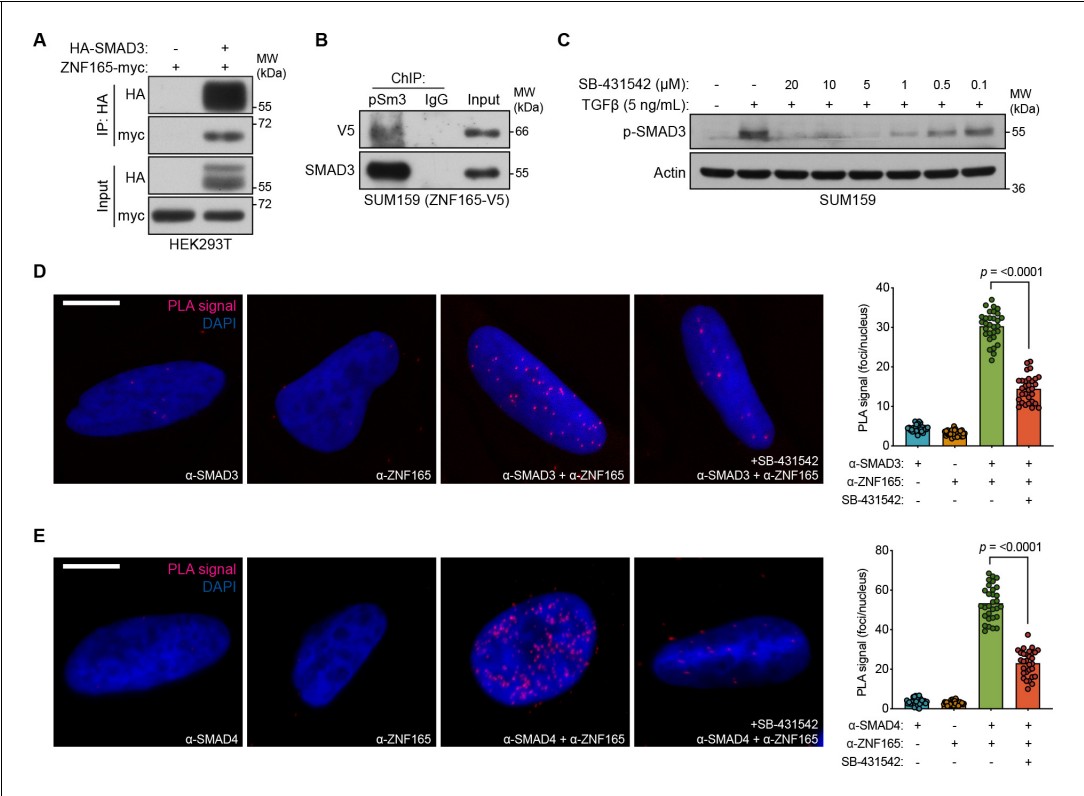

**Figure 3.** ZNF165 physically associates with SMAD3 in a TGFβ-dependent manner. (**A**) Forty-eight hours after transfection with HA-SMAD3 and/or ZNF165-myc cDNA, HEK293T cells were subjected to immunoprecipitation with HA antibody. Immunoblotting was performed with indicated antibodies. Data are representative of two independent experiments. (**B**) ChIP was performed in SUM159 cells stably expressing ZNF165-V5 with antibodies against phosphorylated SMAD3 (Ser423/425) or IgG, and the precipitated material was immunoblotted with indicated antibodies. Data are representative of two independent experiments. (**C**) SUM159 cells were pre-treated with SB-431542 at the indicated concentrations for 15 min, followed by stimulation with 5 ng mL$^{-1}$ TGFβ for 30 min. Immunoblotting was performed with indicated antibodies. Data are representative of two independent experiments. (**D**) Proximity ligation assays (PLAs) performed using antibodies against endogenous ZNF165 and SMAD3 in SUM159 cells, where either antibody alone was used as a negative control. Cells were pre-treated with 20 μM SB-431542 or DMSO for 15 min, followed by stimulation with 5 ng mL$^{-1}$ TGFβ for 30 min. Scale bar, 10 μm. The mean PLA signal (number of foci per nucleus) is quantified (right), where each data point represents the mean signal calculated within one image. P-value was calculated using an unpaired, two-tailed Student's t-test. Ten images were used per condition and data are representative of three independent assays. (**E**) As in (**D**) except antibodies against endogenous ZNF165 and SMAD4 were used.

from its zinc finger array (*Figure 4A*; *Edelstein and Collins, 2005*). We identified five putative inter-actors with SCAN domains and examined the consequences of their depletion on *SMURF2* expression. Of the proteins tested, ZNF446 and SCAND1 phenocopied the changes in gene expression observed for ZNF165 and SMAD3 in two TNBC cell lines (*Figure 4B*). Examination of their tumor expression patterns revealed that ZNF446 is upregulated in breast tumors, an increase that trends with reduced survival in the triple-negative enriched basal-like subtype of breast cancer (HR = 1.89; p=0.059) (*Figure 4—figure supplement 1A and B*; *Perou et al., 2000*; *Sørlie et al., 2001*). More-over, we found that depletion of ZNF446 significantly reduced anchorage-independent growth of TNBC cells (*Figure 4—figure supplement 1C*). Co-expression/co-immunoprecipitation assays con-firmed that ZNF165 and ZNF446 interact and that deletion of the ZNF165 SCAN domain abrogates this association (*Figure 4C*). In addition, an endogenous interaction between ZNF165 and ZNF446 was detectable in TNBC tumor cell nuclear extracts, confirming these proteins interact in the rele-vant subcellular compartment (*Figure 4D*). We then employed the SMAD3/4 PLA assay described above and found that ZNF446 interacts with both SMAD3 and SMAD4 in a TGFβ-dependent fashion, similar to ZNF165 (*Figure 4E and F*).

To determine if ZNF446 also associates with ZNF165-SMAD3 target genes, we performed ChIP-seq for ZNF446-V5 stably expressed in WHIM12 and SUM159 cells. This led to the identification of 5039 and 7926 binding sites in WHIM12 and SUM159 cells, respectively. In both cell lines, ZNF446

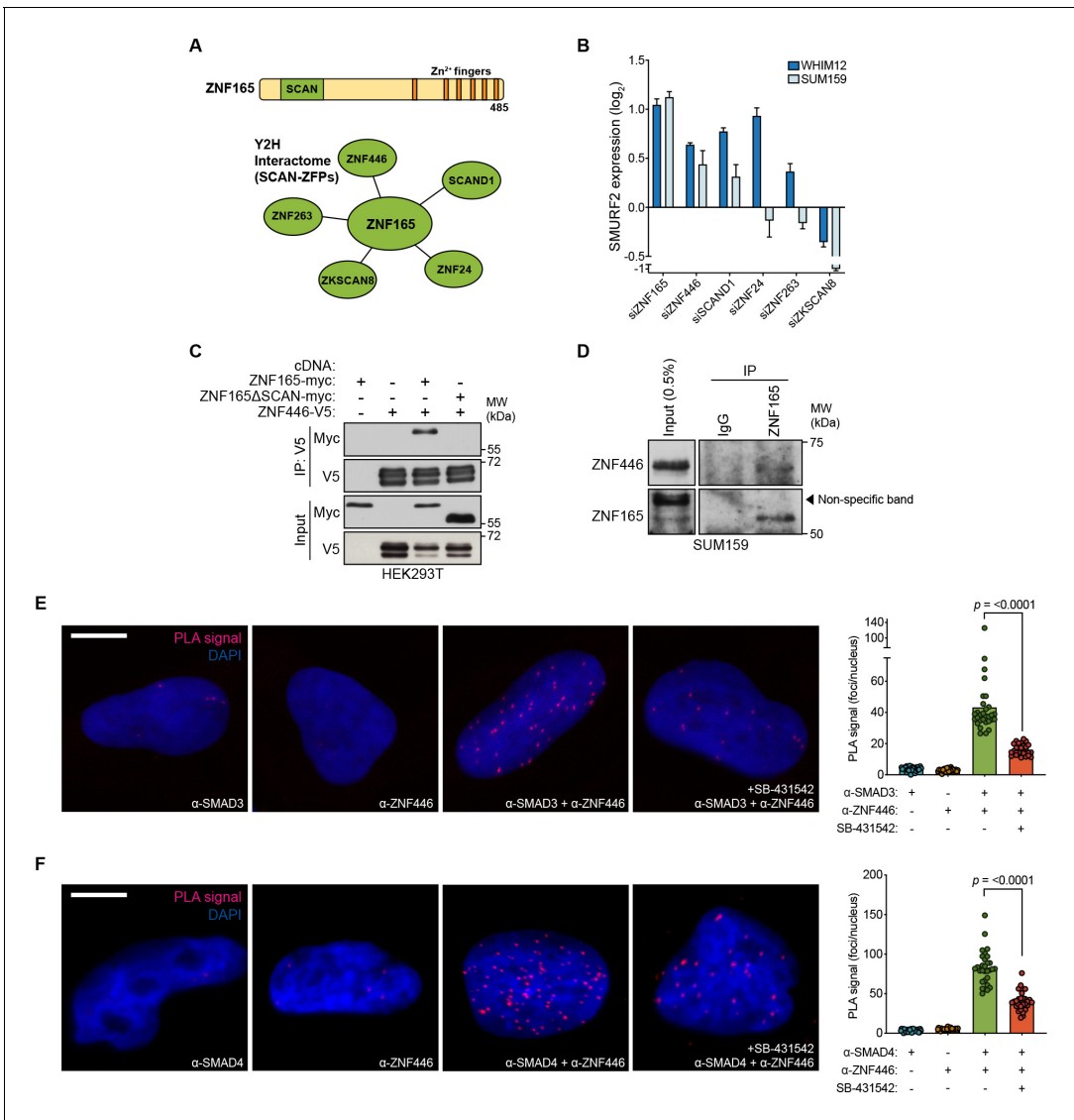

**Figure 4.** ZNF446 is an obligate component of the ZNF165-SMAD3 transcriptional complex. (A) ZNF165 interactors that contain a SCAN domain as identified from a systematic yeast two-hybrid screening approach (interactome.dfci.harvard.edu). A domain map of ZNF165 is shown above for reference. (B) Forty-eight hours after transfection with indicated siRNAs, RNA was extracted from WHIM12 or SUM159 cells and qPCR was used to determine the relative expression of SMURF2. $Log_2$ fold-change values normalized to control knockdown samples are displayed. Error bars represent mean + SEM. P-values were calculated for each sample using a one-tailed Mann-Whitney test. Data are representative of three independent experiments. (C) Forty-eight hours after transfection with indicated cDNA, HEK293T lysates were subjected to immunoprecipitation with V5 antibody. Immunoblotting was performed with indicated antibodies. Data are representative of three independent experiments. (D) SUM159 nuclear lysates were immunoprecipated with antibodies against endogenous ZNF165 or IgG. Immunoblotting was performed with indicated antibodies. Data are representative of two independent experiments. (E) Proximity ligation assays (PLAs) performed using antibodies against endogenous ZNF446 and SMAD3 in SUM159 cells, where either antibody alone was used as a negative control. Cells were pre-treated with 20 µM SB-431542 or DMSO for 15 min, followed by stimulation with 5 ng mL$^{-1}$ TGFβ for 30 min. Scale bar, 10 µm. The mean PLA signal (number of foci per nucleus) is quantified (right), where each data point represents the mean signal calculated within one image. P-value was calculated using an unpaired, two-tailed Student's t-test. Ten images were used per condition and data are representative of three independent assays. (F) As in (E) except antibodies against endogenous ZNF446 and SMAD4 were used.

The online version of this article includes the following figure supplement(s) for figure 4:

**Figure supplement 1.** ZNF446 expression is elevated in breast cancer and associated with poor prognosis.

was enriched at the ZNF165 binding sites (*Figure 5A and B*; *Figure 5—figure supplement 1A–C*). Intersecting the target genes identified for ZNF446 and ZNF165 further revealed a statistically significant overlap in both WHIM12 and SUM159 cells (p<0.0001, hypergeometric distribution) (*Figure 5C and D*). Approximately half of all ZNF446 binding sites in WHIM12 cells were also co-occupied by SMAD3 (*Figure 5E and F*; *Figure 5—figure supplement 1D*). These co-bound sites were largely enriched for promoter regions compared to the total binding sites for either factor (*Figure 5—figure supplement 1E*). Differential motif enrichment analysis revealed the presence of a short GC-rich element at these ZNF446-SMAD3 bound sites, closely resembling the motif we identified at ZNF165-SMAD3 bound sites and identical to the recently reported SMAD consensus binding motif (*Figure 5G and H*; *Figure 1G*; *Martin-Malpartida et al., 2017*). In addition, we found that over half of the TGFβ-responsive target genes shared by ZNF165 and SMAD3 (251/446) are also targeted by ZNF446 (*Figure 5I*). The impact of ZNF446 depletion on expression of these genes significantly correlated with both ZNF165 (r = 0.49) and SMAD3 (r = 0.86) (*Figure 5J*). Together, these findings collectively implicate ZNF446 as a member of the ZNF165-SMAD3 complex that can influence TGFβ-dependent gene regulation in TNBC.

## TRIM27 is essential for ZNF165 transcriptional activity and tumor growth in vivo

ZNF446 contains a KRAB (Krüppel-associated box) domain, which is well documented to interact with the RBCC (RING, B-box, coiled-coil) domain of proteins within the tripartite motif (TRIM) family that in turn function as co-repressors or, as recently demonstrated, co-activators (*Friedman et al., 1996*; *Schultz et al., 2001*; *Schultz et al., 2002*; *Chen et al., 2019*). Y2H-based proteomics confirmed such an interaction of TRIM27, which contains the classic RBCC domain, with ZNF446 (*Figure 6A*; *Supplementary file 3*). Originally identified as an oncogenic fusion with the RET tyrosine kinase, TRIM27 has been ascribed roles in numerous cellular processes that include transcriptional repression, proliferation, meiosis, innate immunity, and endosomal trafficking (*Takahashi et al., 1985*; *Shimono et al., 2000*; *Shimono et al., 2003*; *Zha et al., 2006*; *Hao et al., 2013*; *Zaman et al., 2013*; *Zheng et al., 2015*; *Ma et al., 2016*). However, its function as a transcriptional regulator in TNBC has not been characterized. Co-expression/co-immunoprecipitation experiments revealed that both ZNF446 and ZNF165 interact with TRIM27 and that all three proteins immunoprecipitate as a complex (*Figure 6B*). We subsequently confirmed a nuclear interaction between ZNF165-V5 and endogenous TRIM27 by proximity ligation in SUM159 cells and found that TRIM27 depletion also phenocopied changes in ZNF165-SMAD3 target gene expression (r = 0.81) (*Figure 6C and D*). Notably, TRIM27 appears to be bi-functional, either positively or negatively regulating expression of the ZNF165-SMAD3 targets (quadrant I and III, *Figure 6D*). Indicative of a co-activator function, we found that TRIM27 is required for the transcriptional activation of *RRAD* following ZNF165 or ZNF446 overexpression (*Figure 6E*; *Figure 6—figure supplement 1A*). Similar to ZNF165 and ZNF446, proximity ligation assays revealed endogenous, nuclear interactions between TRIM27 and both SMAD3 and SMAD4, which were reduced by TGFβ pathway inhibition (*Figure 6F and G*).

TRIM27 expression is elevated in several cancers including colon, lung, endometrial and ovarian, and often correlates with poor prognosis (*Tsukamoto et al., 2009*; *Horio et al., 2012*; *Iwakoshi et al., 2012*; *Zoumpoulidou et al., 2012*; *Ma et al., 2016*; *Zhang et al., 2018*). Using TCGA data, we found that TRIM27 expression is significantly elevated in TNBC compared to normal breast tissue (*Figure 6—figure supplement 1B*). Moreover, we stained a small cohort of normal and TNBC tissues for TRIM27 and observed a striking difference in its subcellular localization. TRIM27 appeared to be predominantly cytoplasmic in normal breast epithelia and more nuclear in TNBC tissues, suggesting that its transcriptional activity might associate with the transformed state (*Figure 6—figure supplement 1C*). This finding is in agreement with previous reports and the Human Tissue Atlas, where TRIM27 is mostly cytoplasmic in normal breast and nuclear in testis and a subset of breast cancer tissues (*Tezel et al., 1999*; *Tezel et al., 2009*; *Uhlén et al., 2015*).

Given that TRIM27 was critical for ZNF165-dependent gene regulation, we next asked whether TRIM27 was essential for malignant behaviors. In the absence of extracellular matrix, TRIM27 depletion dramatically reduced tumor cell growth (*Figure 6—figure supplement 1D–F*). We next perturbed TRIM27 as a means to inhibit the activity of the ZNF165 transcriptional complex and assess its impact on TNBC tumor growth and survival at the orthotopic site in mice. Following stable

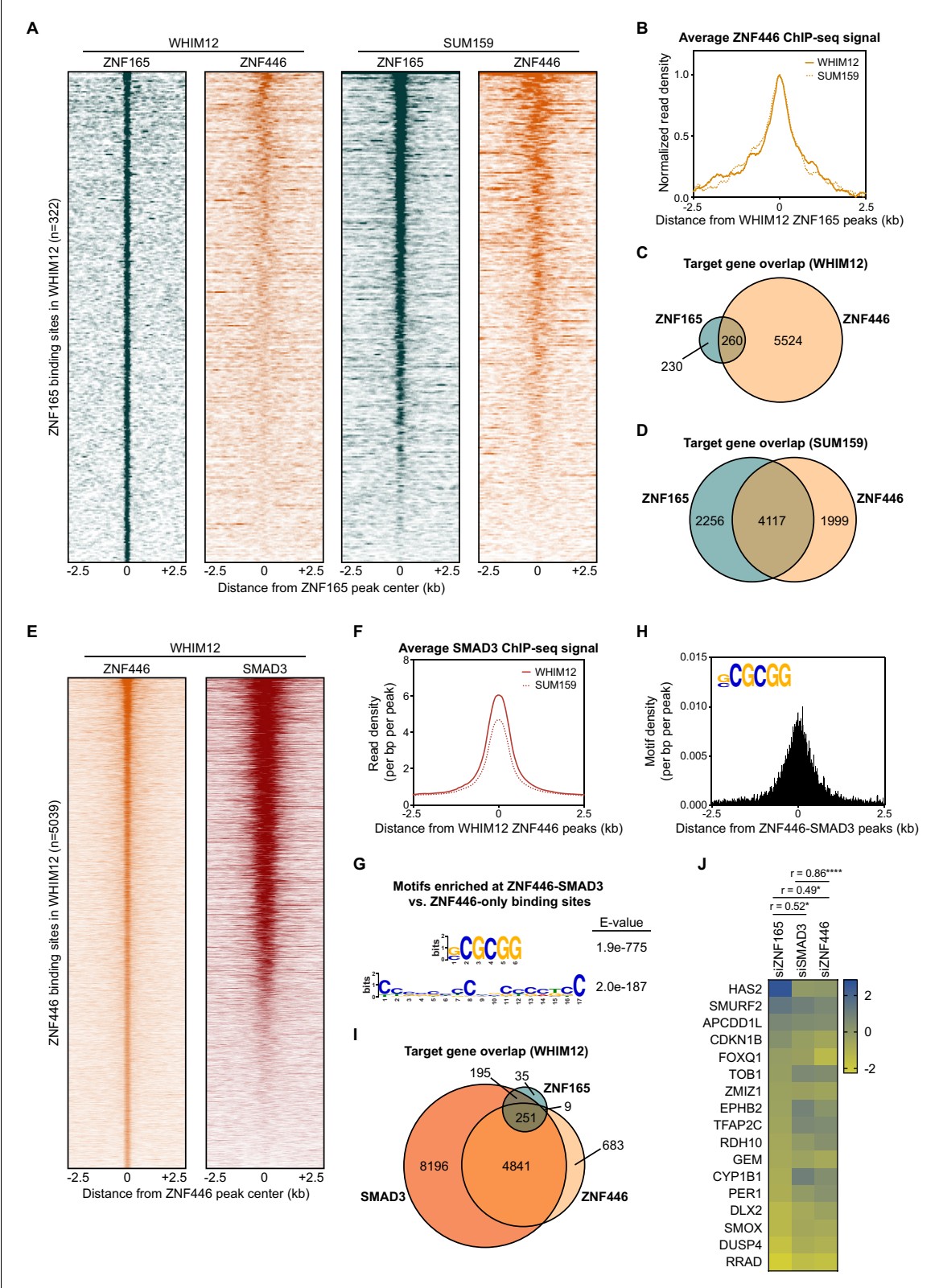

**Figure 5.** ZNF446 co-occupies a subset of ZNF165 and SMAD3 binding sites in TNBC cells. (**A**) Heatmaps of ChIP-seq data for ZNF165 and ZNF446 in the indicated TNBC cell lines. All peaks within each heatmap are centered ±2.5 kb from the WHIM12 ZNF165 peaks (n = 322). (**B**) Normalized read density (per bp per peak) for ZNF446 plotted ±2.5 kb from the ZNF165 peaks identified in WHIM12 cells (n = 322). (**C**) Venn diagram displaying the overlap between ZNF165 and ZNF446 target genes in WHIM12 cells as identified by GREAT (ver 3.0.0). P-value (p<0.0001) was calculated using the

*Figure 5 continued on next page*

*Figure 5 continued*

hypergeometric distribution. (D) As in (C) except using target genes identified for ZNF165 and ZNF446 in SUM159 cells. (E) Heatmaps of ChIP-seq data for ZNF446 and SMAD3 in WHIM12 cells. All peaks within each heatmap are centered ±2.5 kb from the ZNF446 peaks (n = 5,039). (F) Read density (per bp per peak) for SMAD3 plotted ±2.5 kb from the ZNF446 peaks identified in WHIM12 cells (n = 5,039). (G) Motifs enriched at ZNF446-SMAD3 co-bound sites within WHIM12 cells. The ZNF446 binding sites not occupied by SMAD3 (n = 2,587) were used as a control set of sequences to identify differentially enriched motifs at the shared sites (n = 2,376). (H) Motif density (per bp per peak) for the (G|C)CGCGG motif plotted ±2.5 kb from the ZNF446-SMAD3 co-bound peaks in WHIM12 cells. (I) Venn diagram displaying the overlap between ZNF165, SMAD3, and ZNF446 target genes from WHIM12 cells as identified by GREAT (ver 3.0.0). (J) Heatmap displaying $\log_2$ fold-changes of ZNF165-SMAD3 targets in response to depletion of ZNF446, as well as either ZNF165 or SMAD3. The Pearson correlation coefficient is indicated by r. Significance is indicated by asterisks, where *=p < 0.05 and ****=p < 0.0001. Data are representative of four independent experiments.

The online version of this article includes the following figure supplement(s) for figure 5:

**Figure supplement 1.** ZNF446 co-occupies a subset of ZNF165 and SMAD3 binding sites in TNBC cells.

depletion of TRIM27, SUM159 cells were injected into the mammary fat pad of immunocompromised mice (*Figure 6—figure supplement 1E*). We observed a significant decrease in tumor size, suggesting that TRIM27 is essential for TNBC tumor growth in vivo and that inhibition of the ZNF165 transcriptional complex may be a promising avenue of therapeutic intervention (*Figure 6H and I*). Together, these results indicate that TRIM27 is a previously unrecognized regulatory component of TGFβ signaling in TNBC and functions to modulate TGFβ-dependent transcription in cooperation with ZNF165, ZNF446, and SMAD3.

## Discussion

Striking similarities between tumors and developing germ cells have been noted for over a century (*Simpson et al., 2005*). As a result, it has been speculated that a fundamental attribute of cancer is the usurpation of developmental signaling programs. Tumors frequently activate expression of gametogenic genes, which encode a group of proteins termed cancer/testis antigens (CT antigens) (*Chen et al., 1997*; *Simpson et al., 2005*). While these proteins have long been considered optimal targets for immunotherapy, their function, if any, in tumor cells has not been well-established.

In recent years, an increasing number of studies have begun to indicate that CT antigens are not merely innocuous by-products of aberrant gene expression programs, but instead can be essential to the progression of cancer. The list of tumor cell mechanisms regulated by these proteins includes mitosis, protein turnover, mitigating DNA damage, and disrupting normal transcriptional programs (*Gibbs and Whitehurst, 2018*). This variety of biological processes is attributable to the high degree of diversity in the functions of proteins classified as CT antigens, and speaks to the ability of tumor cells to engage aberrant gene expression programs in order to support their growth and survival. In support of this, our study of the testis protein ZNF165 has revealed a previously unrecognized mechanism by which TGFβ signaling is co-opted to promote malignancy in late-stage breast tumors.

Our findings expose a high degree of flexibility in tumor cell transcriptional networks. We reason that this feature could be the result of the modular domains present in many transcriptional regulators, particularly zinc finger proteins (ZFPs), which make up the largest family of transcription factors in the human genome (*Lambert et al., 2018*). The vast majority of these proteins contain KRAB, BTB/POZ, or SCAN domains, that mediate interactions with transcriptional partners (*Bellefroid et al., 1991*; *Bardwell and Treisman, 1994*; *Albagli et al., 1995*; *Williams et al., 1995*). The SCAN domain in particular regulates the assembly and thus function of specific transcriptional complexes (*Edelstein and Collins, 2005*). SCAN domains are not promiscuous, but demonstrate selectivity for co-association, the basis of which is not clear. Nevertheless, ectopic introduction of a SCAN domain-containing protein, such as ZNF165, may drive assembly of a neomorphic transcriptional complex, ZNF165-ZNF446-SMAD3-TRIM27, that is otherwise not present in somatic cells (*Figure 7*). This finding highlights the importance of silencing lineage-specific transcription factors, as their anomalous expression may be sufficient to disrupt transcriptional networks that regulate tissue homeostasis.

The study of CT antigen function generally, and ZNF165 in particular, has the capacity to reveal new principles underlying tumor cell regulatory networks. Specifically, our data indicate that both ZNF446 and TRIM27 are complicit in the activation of oncogenic TGFβ signaling in TNBC. While

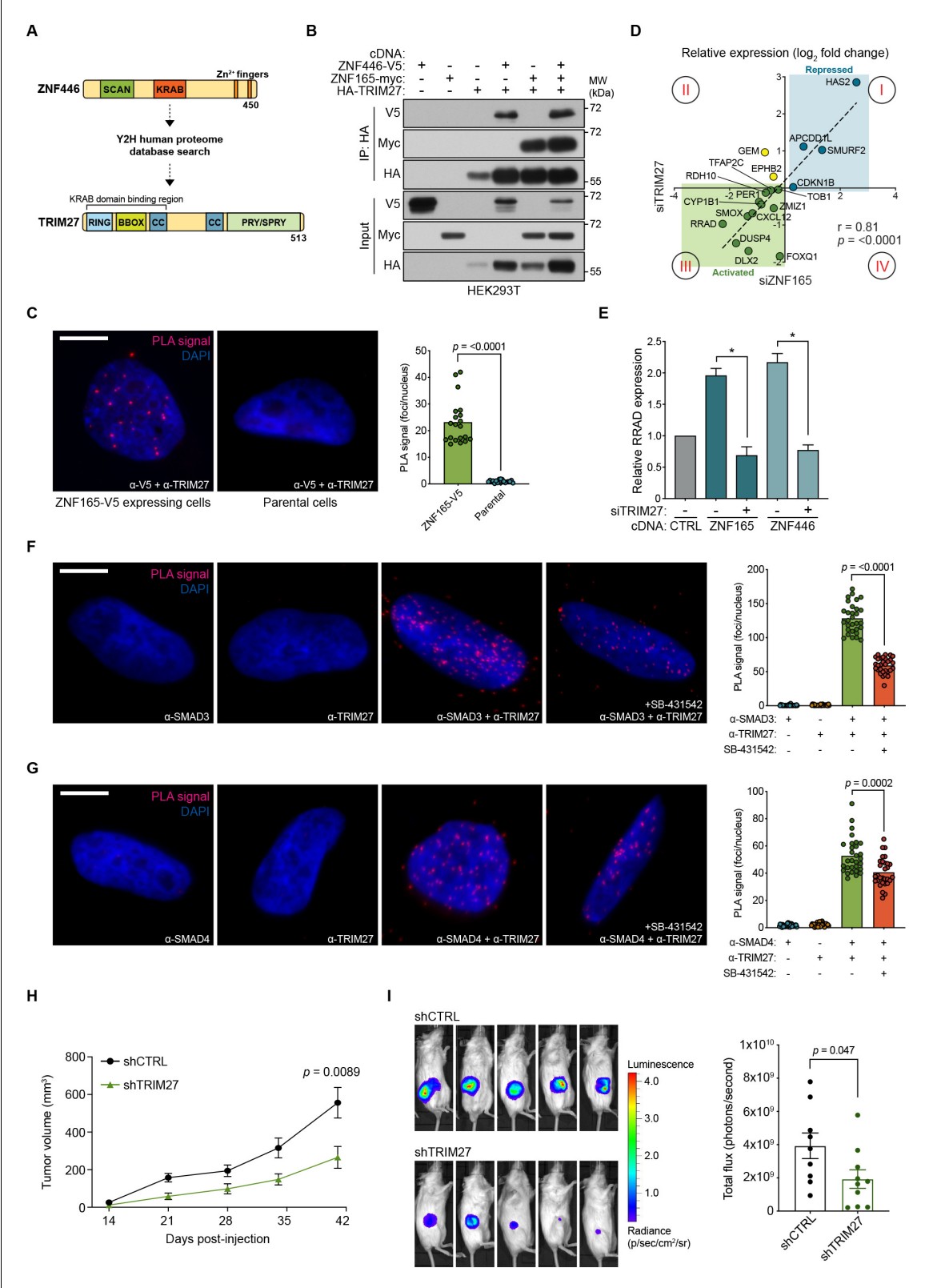

**Figure 6.** TRIM27 is essential for ZNF165 transcriptional activity and tumor growth in vivo. (**A**) Cartoon maps showing the domains of ZNF446 and TRIM27. The KRAB-binding domain of TRIM27 is highlighted for reference. (**B**) Forty-eight hours after transfection with indicated cDNA, HEK293T lysates were subjected to immunoprecipitation with HA antibody. Immunoblotting was performed using indicated antibodies. Data are representative of three independent experiments. (**C**) Proximity ligation assays (PLAs) performed using V5 and TRIM27 antibodies in SUM159 cells stably expressing

*Figure 6 continued on next page*

*Figure 6 continued*

ZNF165-V5. Parental SUM159 cells were used as a negative control with the V5/TRIM27 antibody combination. Scale bar, 10 μm. The mean PLA signal (number of foci per nucleus) is quantified (right), where each data point represents the mean signal calculated within one image. P-values were calculated using an unpaired, two-tailed Student's t-test. Five images were used per condition and data are representative of four independent assays. (D) WHIM12 cells were transfected with siRNA for 48 hr and qPCR was used to quantify relative expression (log$_2$ fold change) of shared ZNF165-SMAD3 target genes upon depletion of ZNF165 (x-axis) or TRIM27 (y-axis). The Pearson correlation coefficient is indicated by r. Data are representative of three independent experiments. (E) SUM159 cells stably expressing indicated cDNA were transfected with siRNA targeting TRIM27 for 48 hr. Relative RRAD expression was measured using qPCR and the data were normalized to the CTRL sample (grey). Error bars represent mean + SEM. P-values were calculated using an unpaired, two-tailed Mann-Whitney test. Significance is indicated by asterisks, where *=p < 0.05. Data are representative of four independent experiments. (F) PLAs performed using antibodies against endogenous TRIM27 and SMAD3 in SUM159 cells, where either antibody alone was used as a negative control. Cells were pre-treated with 20 μM SB-431542 or DMSO for 15 min, followed by stimulation with 5 ng mL$^{-1}$ TGFβ for 30 min. Scale bar, 10 μm. The mean PLA signal (number of foci per nucleus) is quantified (right), where each data point represents the mean signal calculated within one image. P-value was calculated using an unpaired, two-tailed Student's t-test. Ten images were used per condition and data are representative of three independent assays. (G) As in (F) except antibodies against endogenous TRIM27 and SMAD4 were used. (H) Tumor volumes from mice orthotopically injected with SUM159T-Luciferase cells stably expressing shRNAs against TRIM27 (n = 10) or a non-targeting control (n = 9). P-value was calculated using an unpaired, two-tailed Student's t-test. (I) Representative images of bioluminescence (BLI) measurements taken for mice from each group. BLI data was quantified (right) and the p-value was calculated using an unpaired, two-tailed Student's t-test.

The online version of this article includes the following figure supplement(s) for figure 6:

**Figure supplement 1.** Analysis of TRIM27 expression and contribution to neoplastic behaviors in TNBC.

ZNF446 has been reported to act as a transcriptional repressor for SRF and AP-1 via its KRAB domain, a physiological function for ZNF446 in either normal or malignant contexts has not been reported (*Liu et al., 2005*). Our findings suggest that in addition to functioning as a chromatin-bound cofactor for ZNF165 and SMAD3, ZNF446 may recruit TRIM27 to modulate TGFβ-induced transcription in TNBC. An important caveat to this conclusion is that the transcriptional effects we observed are based on changes in steady-state mRNA levels, a fraction of which could conceivably be due to post-transcriptional effects, a possibility that we did not fully rule out. Regardless, we find that ZNF446 and TRIM27 can either activate or repress expression of genes targeted by the

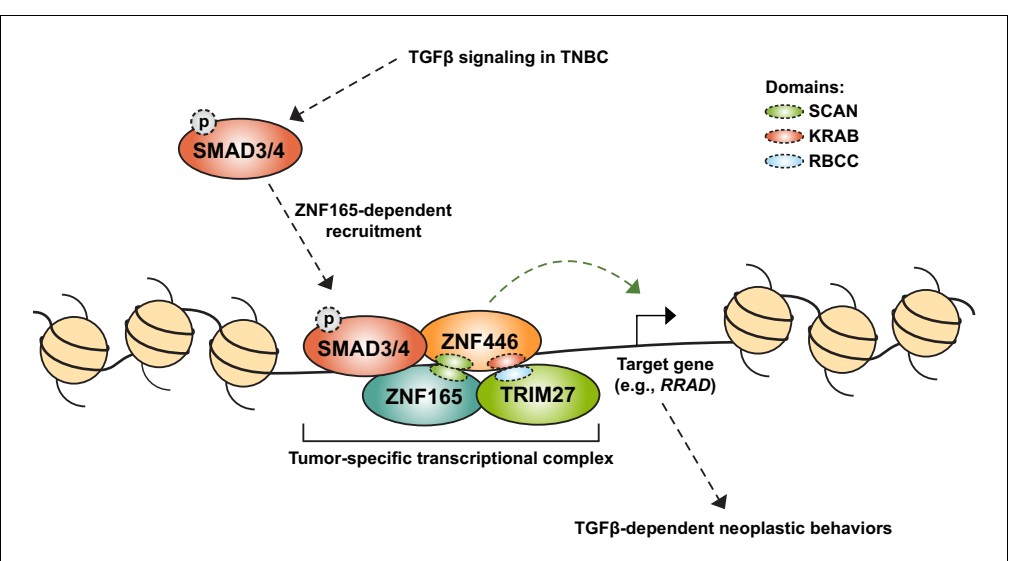

**Figure 7.** Model for assembly of a neomorphic ZNF165 transcriptional complex in TNBC. Aberrant expression of testis-specific ZNF165 in TNBC co-opts somatic transcriptional machinery to modulate TGFβ-dependent transcription. The ZNF165 SCAN domain mediates interaction with ZNF446, and together these factors associate with SMAD3 at shared binding sites throughout the genome and alter gene expression to promote neoplastic behaviors. Via its KRAB domain, ZNF446 likely recruits the co-activator/co-repressor TRIM27, which is essential for ZNF165 transcriptional activity and tumor growth in vivo. Given the testis/tumor-restricted expression of ZNF165, disruption of this complex holds potential as a therapeutic strategy to specifically inhibit pro-tumorigenic TGFβ signaling in TNBC.

ZNF165-SMAD3 complex, despite reports describing only repressive capabilities for these factors (*Shimono et al., 2000*; *Shimono et al., 2003*; *Liu et al., 2005*). This finding also dovetails with a recent report demonstrating that TRIM28 serves as a co-activator for ZFP30-mediated adipogenesis, and together these studies begin to establish that tripartite motif proteins may be bi-functional with respect to transcriptional repression or activation (*Chen et al., 2019*). Notably, TRIM27 is also a well characterized E3 ligase and has been reported to modify numerous protein substrates via ubiquitination and SUMOylation (*Chu and Yang, 2011*; *Zurek et al., 2012*; *Lee et al., 2013*). Future studies to address whether the catalytic activity of TRIM27 is required for ZNF165-dependent transcriptional regulation are therefore crucial, as small molecule inhibitors of TRIM27 could selectively disrupt pro-tumorigenic TGFβ signaling in TNBC.

An additional, unexpected finding is that *RRAD*, whose transcription is highly regulated by the ZNF165 transcriptional complex, is a TGFβ target gene that can exert opposing behaviors depending on context. TGFβ signaling is well known to drive pro-tumorigenic phenotypes in less differentiated breast tumor cell types, such as TNBC, while facilitating tumor suppression in more differentiated cell types (*Tian et al., 2003*; *Massagué, 2008*). Our study indicates that RRAD contributes to such context-specific phenotypes conferred by TGFβ signaling, which may help to explain extensive data that report opposing impacts of RRAD expression in a variety of cancer types (*Suzuki et al., 2007*; *Lee et al., 2010*; *Mo et al., 2012*; *Yeom et al., 2012*). With respect to function, RRAD is a small GTPase reported to negatively regulate L-type calcium channels in the heart, where its deficiency can promote cardiac hypertrophy (*Chang et al., 2007*; *Manning et al., 2013*; *Meza et al., 2013*). In cancer, RRAD has been implicated in numerous processes, including the regulation of glycolysis, proliferation, NF-κB signaling, and apoptosis (*Lee et al., 2010*; *Yeom et al., 2012*; *Hsiao et al., 2014*; *Zhang et al., 2014*; *Liu et al., 2015*; *Shang et al., 2016*). Although the exact function of RRAD in TNBC will need to be elucidated in future studies, the vast array of phenotypes associated with this protein undoubtedly reflect the context within which it functions. Ultimately, further investigation could indicate that RRAD itself is a tractable intervention point in TNBC.

# Materials and methods

**Key resources table**

| Reagent type (species) or resource | Designation | Source or reference | Identifiers | Additional information |
|---|---|---|---|---|
| Strain, strain background (*Mus musculus*) | NOD.Cg-*Prkdc*$^{scid}$ *Il2rg*$^{tm1Wjl}$/SzJ; NSG | The Jackson Laboratory | Stock no. 005557; RRID:IMSR_JAX:005557 | |
| Cell line (*Homo sapiens*) | HEK293T | ATCC | CRL-3216; RRID:CVCL_0063 | |
| Cell line (*H. sapiens*) | MCF7 | ATCC | HTB-22; RRID:CVCL_0031 | |
| Cell line (*H. sapiens*) | WHIM12 | *Li et al., 2013* | | |
| Cell line (*H. sapiens*) | SUM159 | Asterand | RRID:CVCL_5423 | |
| Cell line (*H. sapiens*) | SUM159T-Luciferase | *Westcott et al., 2015* | | |
| Antibody | anti-HA (Rat monoclonal) | Roche | Cat #11867423001; RRID:AB_390918 | IB (1:1000), IP (1 μg) |
| Antibody | anti-c-Myc (Rabbit polyclonal) | Santa Cruz Biotechnology | sc-789; RRID:AB_631274 | IB (1:1000) |
| Antibody | anti-V5 (Rabbit monoclonal) | Cell Signaling Technology | Cat #13202; RRID:AB_2687461 | IB (1:1000) |

*Continued on next page*

*Continued*

| Reagent type (species) or resource | Designation | Source or reference | Identifiers | Additional information |
|---|---|---|---|---|
| Antibody | anti-SMAD3 (Rabbit polyclonal) | Abcam | ab28379; RRID:AB_2192903 | IB (1:1000), PLA (1:200), ChIP-qPCR (1.5 µg), ChIP-seq (10 µg) |
| Antibody | anti-phospho-SMAD3 (Rabbit monoclonal) | Cell Signaling Technology | Cat #9520; RRID:AB_2193207 | IB (1:1000), ChIP (5 µg) |
| Antibody | anti-SMURF2 (Mouse monoclonal) | Santa Cruz Biotechnology | sc-393848 | IB (1:500) |
| Antibody | anti-RRAD (Goat polyclonal) | Thermo Fisher Scientific | PA537885; RRID:AB_2554493 | IB (1:1000) |
| Antibody | anti-ZNF165 (Mouse monoclonal) | Novus Biologicals | H00007718-M02; RRID:AB_1717273 | IB (1:1000) |
| Antibody | anti-ZNF165 (Rabbit polyclonal) | Atlas Antibodies | HPA007247; RRID:AB_2797217 | IB (1:1000), PLA (1:200), IP (5 µg) |
| Antibody | anti-Tubulin (Rabbit monoclonal) | Cell Signaling Technology | Cat #2128; RRID:AB_823664 | IB (1:1000) |
| Antibody | anti-ZNF446 (Rabbit polyclonal) | Proteintech | 16218–1-AP | IB (1:1000), PLA (1:200) |
| Antibody | anti-TRIM27 (Rabbit monoclonal) | Cell Signaling Technology | Cat #15099; RRID:AB_2798707 | IB (1:1000) |
| Antibody | anti-V5 (Mouse monoclonal) | Thermo Fisher Scientific | R960-25; RRID:AB_2556564 | PLA (1:500), IP (1 µg) |
| Antibody | anti-SMAD3 (Mouse monoclonal) | Santa Cruz Biotechnology | sc-101154; RRID:AB_1129525 | PLA (1:200) |
| Antibody | anti-SMAD4 (Mouse monoclonal) | Santa Cruz Biotechnology | sc-7966; RRID:AB_627905 | PLA (1:200) |
| Antibody | anti-TRIM27 (Rabbit polyclonal) | Proteintech | 12205–1-AP; RRID:AB_2256660 | PLA (1:200), IHC (1:200) |
| Antibody | anti-H3K27ac (Rabbit polyclonal) | Abcam | ab4729; RRID:AB_2118291 | ChIP-qPCR (2 µg) |
| Antibody | Normal rabbit IgG (Rabbit polyclonal) | Cell Signaling Technology | Cat #2729; RRID:AB_1031062 | IP (1 µg), ChIP-qPCR (1.5–5 µg) |
| Antibody | anti-V5 (Rabbit polyclonal) | Abcam | ab9116; RRID:AB_307024 | ChIP-seq (5 µg) |
| Peptide, recombinant protein | Human Transforming Growth Factor-β1 | Cell Signaling Technology | Cat #8915 | |
| Chemical compound, drug | SB-431542 | Tocris Bioscience | Cat #1614 | |
| Commercial assay or kit | KAPA HyperPrep Kit | KAPA Biosystems | KK8502 | |
| Commercial assay or kit | Duolink In Situ PLA Probe Anti-Mouse PLUS | Sigma-Aldrich | DUO92001; RRID:AB_2810939 | |
| Commercial assay or kit | Duolink In Situ PLA Probe Anti-Rabbit MINUS | Sigma-Aldrich | DUO92005; RRID:AB_2810942 | |

*Continued*

| Reagent type (species) or resource | Designation | Source or reference | Identifiers | Additional information |
|---|---|---|---|---|
| Commercial assay or kit | Duolink In Situ Detection Reagent Red | Sigma-Aldrich | DUO92008 | |
| Software, algorithm | ImageJ (Fiji) | *Schindelin et al., 2012* | RRID:SCR_002285 | ver 2.0.0 |
| Software, algorithm | Bowtie2 | *Langmead and Salzberg, 2012* | RRID:SCR_005476 | ver 2.3.2 |
| Software, algorithm | SAMtools | *Li et al., 2009* | RRID:SCR_002105 | ver 1.6 |
| Software, algorithm | DeepTools | *Ramírez et al., 2016* | RRID:SCR_016366 | ver 2.3.5 |
| Software, algorithm | Integrative Genomics Viewer | *Robinson et al., 2011* | RRID:SCR_011793 | ver 2.3.93 |
| Software, algorithm | HOMER | *Heinz et al., 2010* | RRID:SCR_010881 | ver 4.9 |
| Software, algorithm | GREAT | *McLean et al., 2010* | RRID:SCR_005807 | ver 3.0.0 |
| Software, algorithm | MEME | *Bailey et al., 2009* | RRID:SCR_001783 | ver 5.1.1 |
| Software, algorithm | GSEA | *Subramanian et al., 2005* | RRID:SCR_003199 | ver 4.0.3 |

## Cell lines and transfections

Cell lines were obtained from American Type Culture Collection (ATCC) except for: WHIM12 (Matthew Ellis, Baylor College of Medicine), SUM159 (Asterand), and SUM159T-Luceiferase (Gray Pearson, Georgetown University). All cell lines were cultured in the provider's recommended medium and authenticated using short-tandem repeat (STR) profiling. Cell lines were regularly tested for mycoplasma and no contamination was observed. For siRNA transfections, cells were trypsinized and seeded in Opti-MEM containing Lipofectamine RNAiMAX (Thermo Fisher Scientific, Waltham, MA) and pooled siRNAs at a final concentration of 50 nM. cDNA transfections in HEK293T cells were performed using the calcium phosphate method as previously described (*Maxfield et al., 2015*).

## Antibodies and reagents

The following antibodies were used for immunoblotting: HA (3F10; 1:1000; Roche, Basel, Switzerland), c-Myc (sc-789; 1:1000; Santa Cruz Biotechnology; Dallas, TX), V5 (13202; 1:1000; Cell Signaling Technology, Danvers, MA), SMAD3 (ab28379; 1:1000; Abcam, Cambridge, United Kingdom), phospho-SMAD3 (9520, 1:1000; Cell Signaling Technology), SMURF2 (sc-393848; 1:500; Santa Cruz Biotechnology), RRAD (PA537885; 1:1000; Thermo Fisher Scientific), ZNF165 (H00007718-M02; 1:1000; Novus Biologicals, Centennial, CO), ZNF165 (HPA007247, 1:1000; Atlas Antibodies, Bromma, Sweden), Tubulin (2128; 1:1000; Cell Signaling Technology), ZNF446 (16218–1-AP; 1:1000; Proteintech, Rosemont, IL), TRIM27 (15099; 1:1000; Cell Signaling Technology). The following antibodies were used for proximity ligation assays: V5 (R960-25; 1:500; Thermo Fisher Scientific), SMAD3 (ab28379; 1:200; Abcam), SMAD3 (sc-101154, 1:200; Santa Cruz Biotechnology), SMAD4 (sc-7966; 1:200, Santa Cruz Biotechnology), ZNF165 (HPA007247, 1:200; Atlas Antibodies), ZNF446 (16218–1-AP; 1:200; Proteintech), TRIM27 (12205–1-AP; 1:200, Proteintech). ChIP-qPCR experiments were performed with 1.5 µg SMAD3 (ab28379; Abcam; Lot no. GR321957-8), 5 µg phospho-SMAD3 (9520; Cell Signaling Technology; Lot no. 15), 2 µg H3K27ac (ab4729; Abcam; Lot no. GR323132-1), or the appropriate amount of normal rabbit IgG (2729; Cell Signaling Technology; Lot no. 9). ChIP-seq in WHIM12 and SUM159 cells was performed with 10 µg SMAD3 (ab28379; Abcam; Lot no. GR321957-8) or 5 µg V5 (ab9116; Abcam; Lot no. GR3215823-1). Immunoprecipitations used for interaction studies were performed with HA (3F10; Roche), V5 (R960-25; Thermo Fisher Scientific),

and ZNF165 (HPA007247; Atlas Antibodies). IHC was performed using TRIM27 (12205–1-AP; Proteintech) at 1:200.

Transforming growth factor-β1 was purchased from Cell Signaling Technology (8915) and SB-431542 was purchased from Tocris Bioscience (Bristol, United Kingdom) (1614). siRNAs targeting ZNF165, SMAD3, ZNF446, TRIM27, SCAND1, ZNF24, ZNF263, or ZKSCAN8, and non-targeting control #2 were obtained from Dharmacon (Lafayette, CO) (*Supplementary file 4*). siRNAs targeting RRAD (SASI_Hs01_00221658, SASI_Hs01_00221655, SASI_Hs01_00221652, SASI_Hs01_00221653) and shRNAs targeting TRIM27 (SHCLNG-NM_006510; TRCN0000280319 and TRCN0000011021) or a non-targeting control were purchased from Sigma (St. Louis, MO) (*Supplementary file 4*).

## Expression plasmids

ZNF165 and ZNF446 cDNA was purchased in pDONR223 vectors from the DNASU Plasmid Repository (Tempe, AZ) and cloned into pLX302-V5 or pLX304-V5 lentiviral vectors, respectively, using the Gateway Cloning system (Thermo Fisher Scientific). psPAX2 and pMD2.G were gifts from Didier Trono (Addgene plasmids #12260 and #12259). SMAD3, SMAD4, and TRIM27 cDNA was obtained in pENTR221 vectors from the UT Southwestern (UTSW) McDermott Center for Human Genetics and subcloned into pCMV-HA or pCMV-myc (Clontech, Mountain View, CA) between SalI and NotI restriction sites. pCMV6-ZNF165-myc was obtained from Origene (Rockville, MD) (RC205600). The SCAN domain of pCMV6-ZNF165-myc was deleted using overlap extension PCR and verified by sequencing and western blot (primers listed in *Supplementary file 4*).

## Generation of stable cell lines

Stable cell lines were generated through lentiviral-mediated transduction. HEK293T cells were co-transfected with the target gene vectors (pLX302 or pLX304) and lentiviral packaging plasmids (psPAX2 and pMD2.G). Virus conditioned media was collected 48 hr post transfection and used to infect target cells in the presence of polybrene. Selection with appropriate antibiotics was performed for 3 days following infection to generate stable populations. ZNF165-V5 or ZNF446-V5 expression was confirmed by RT-qPCR, western blotting, and immunofluorescence.

## Western blotting and gene expression analysis

Western blotting was done as previously described (*Maxfield et al., 2015*). RNA was isolated using the GenElute Mammalian Total RNA Miniprep Kit (Sigma), and 500 ng RNA was reverse transcribed using the High Capacity cDNA Reverse Transcription Kit (Thermo Fisher Scientific) according to manufacturer's instructions. Taqman Real-Time PCR (Thermo Fisher Scientific) gene expression assays were run in technical duplicate to quantify expression on an Applied Biosystems QuantStudio 5 Real-Time PCR System. Relative expression values were calculated using the $2^{-\Delta\Delta CT}$ method and the data were normalized to RPL27 values. Catalog numbers for Taqman probes used to measure gene expression can be found in *Supplementary file 4*.

## Chromatin immunoprecipitation (ChIP) and sequencing (ChIP-seq)

Cells were grown to ~90% confluency and crosslinked with 1% formaldehyde for 15 min at 37°C. Crosslinking was then quenched with 0.125 M glycine for 5 min at 4°C. Cells were washed twice with cold PBS and lysed in hypotonic buffer (5 mM PIPES pH 8, 85 mM KCl, 0.5% NP-40, 1 mM DTT). All buffers were supplemented with Complete Protease Inhibitor Cocktail (Sigma). Nuclei were pelleted at 2,000 rpm for 5 min and resuspended in lysis buffer containing 50 mM Tris pH 7.9, 1% SDS, 10 mM EDTA, and 1 mM DTT. Chromatin was sheared into 200–400 bp fragments using a Bioruptor sonicator (Diagenode), clarified by centrifugation at max speed for 10 min, and diluted (1:10) in buffer containing 20 mM Tris pH 7.9, 0.5% Triton X-100, 150 mM NaCl, 2 mM EDTA, and 1 mM DTT. Lysates were pre-cleared with Protein A/G agarose beads (Thermo Fisher Scientific) for 1 hr and, following removal of input material, incubated with appropriate antibodies overnight at 4°C. Protein A/G agarose beads blocked with BSA were then added to lysates for 2 hr, followed by one low salt wash (20 mM Tris pH 7.9, 125 mM NaCl, 0.05% SDS, 1% Triton X-100, 2 mM EDTA), one high salt wash (20 mM Tris pH 7.9, 500 mM NaCl, 0.05% SDS, 1% Triton X-100, 2 mM EDTA), a more stringent wash (10 mM Tris pH 7.9, 250 mM LiCl, 1% NP-40, 1% sodium deoxycholate, 1 mM EDTA), and a final wash in TE buffer (10 mM Tris pH 7.9, 1 mM EDTA). For western blotting of

immunoprecipitated proteins, beads were boiled in 2X Laemmli sample buffer (containing 4% β-mer-captoethanol) for 5–10 min. Otherwise, chromatin was eluted from beads with 100 mM NaHCO$_3$ and 1% SDS, and de-crosslinking was carried out overnight at 65˚C. RNA and protein were digested with RNase A (3 µg/mL) and Proteinase K (50 µg/mL), respectively. DNA was purified using phenol/chloroform/isoamyl alcohol (25:24:1) and precipitated overnight with isopropyl alcohol at −20˚C.

Libraries of ChIP DNA were prepared using the KAPA Hyper Prep Kit (KAPA Biosystems, Cape Town, South Africa) according to the manufacturer's instructions. Samples were multiplexed with Illumina adapters (NEB, Ipswich, MA) and 75 bp single end reads (~50M per sample) were generated on an Illumina NextSeq 500 (UTSW McDermott Center Next Generation Sequencing Core). Raw reads were aligned to the human genome assembly (hg19) using Bowtie2 (ver 2.3.2) with the 'sensitive' parameters enabled, and the mapped read files were sorted and indexed with SAMtools (ver 1.6). BigWig files for visualization on the IGV genome browser (*Robinson et al., 2011*) were generated using DeepTools (ver 2.3.5). ChIP-seq was performed using biological duplicates for each sample. WHIM12 ZNF165-V5 ChIP-seq data was previously generated and is available under GEO accession GSE65937. All other ChIP-seq datasets have been deposited under GEO accession GSE130364.

## ChIP-seq peak calling and annotation

Peaks were called using the HOMER (ver 4.9) findPeaks module with size and minimum distance parameters set to 150 bp and 370 bp, respectively. Target genes for each transcription factor were identified using GREAT version 3.0.0 (great.stanford.edu) with default parameters. The HOMER mergePeaks module was used to identify overlapping peaks with a size parameter of 1000 bp. Tag density plots were generated using the HOMER annotatePeaks module with size and bin size parameters set to 5000 and 5 bp, respectively. The same module with the genomeOntology parameter enabled was used to annotate the genomic regions associated with each factor. For heatmap generation, the HOMER annotatePeaks module was used to create matrix files, which were clustered with Cluster 3.0 and visualized with Java TreeView (ver 1.1.6r4). To identify motifs enriched at co-bound sites, motif discovery was performed with MEME (ver 5.1.1) in 'differential enrichment' mode. Motif density plots were generated using the HOMER annotatePeaks module with size and bin size parameters set to 5000 and 5 bp, respectively.

## ChIP-qPCR

ChIP was performed as described above, and qPCR was carried out using SYBR green reagents in technical duplicates (Thermo Fisher Scientific) on an ABI QuantStudio 5 Real-Time PCR System. Three ten-fold dilutions were used to generate standard curves and the amount of ChIP DNA relative to corresponding input values (percent input) was calculated. SMAD3 enrichment was determined relative to IgG percent input values within the same treatment or condition. Primer sequences are listed in *Supplementary file 4*.

## Proximity ligation assays

Proximity ligation assays (PLAs) were performed using Duolink reagents (Sigma) according to manufacturer's instructions. Briefly, cells attached to glass coverslips were fixed with 3.7% formaldehyde for 15 min and then permeabilized with 0.5% Triton X-100 for 15 min. Blocking was performed for 1 hr at 37˚C, and cells were incubated with appropriate antibodies overnight at 4˚C. After washing, coverslips were incubated with PLUS (rabbit) and MINUS (mouse) secondary probes for 1 hr at 37˚C. Ligation was performed for 30 min at 37˚C, followed by rolling circle amplification for 100 min at 37˚C. Coverslips were washed, mounted to glass slides in medium containing DAPI, and kept at 4˚C. Cells were imaged using a Keyence BZ-X710 fluorescence microscope at 100x magnification and 5–10 regions of interest were captured on each coverslip. PLA signals were quantified using ImageJ by counting the average number of foci per nucleus in any given image.

## Immunoprecipitation

For co-expression/co-immunoprecipitation studies, cells were lysed for 30 min on ice in non-denaturing lysis buffer (NDLB) containing 50 mM HEPES pH 7.4, 150 mM NaCl, 1% Triton X-100, 0.5% sodium deoxycholate, 25 mM β-glycerophosphate, 1 mM EDTA, 1 mM EGTA, 1 mM Na$_3$VO$_4$, 1 µg

mL$^{-1}$ pepstatin, 2 µg mL$^{-1}$ leupeptin, 2 µg mL$^{-1}$ aprotinin, and 10 µM bestatin. Lysates were clarified at 12,000 g for 10 min and pre-cleared with Protein A/G agarose beads for 1 hr at 4°C. Five percent of each lysate was set aside as input material and the remainder was immunoprecipitated with appropriate antibodies for 0.5–4 hr at 4°C. Protein A/G agarose beads blocked with 0.5% BSA were added in the final 30 min of immunoprecipitation. Beads were then washed three times in NDLB and once in NDLB containing 500 mM NaCl. For endogenous immunoprecipitation of ZNF165, ~2×10$^8$ cells were collected and lysed in a hypotonic buffer (5 mM PIPES pH 8, 85 mM KCl, 0.5% NP-40) supplemented with Complete Protease Inhibitor Cocktail (Sigma) for 20 min. Nuclei were pelleted at 2,000 rpm for 5 min and resuspended in RIPA buffer (50 mM HEPES pH 7.4, 150 mM NaCl, 1% Triton X-100, 0.5% sodium deoxycholate, 0.5% SDS, 25 mM β-glycerophosphate, 1 mM EDTA, 1 mM EGTA, 1 mM Na$_3$VO$_4$) supplemented with Complete Protease Inhibitor Cocktail. Nuclear lysis was facilitated by gentle agitation at 4°C with intermittent sonication (5–10 pulses at 5 s each using a Fisher Scientific Model 60 Sonic Dismembrator). Lysates were clarified at 16,000 g for 10 min, concentrated to a volume of 1–2 mL, and pre-cleared with Protein A/G agarose beads for 1 hr at 4°C. Five percent of each lysate was set aside as input material and the remainder was subjected to immunoprecipitation overnight at 4°C. The next day, Protein A/G agarose beads blocked with BSA were added to lysates for 2 hr, followed by four washes in a buffer containing 50 mM HEPES pH 7.4, 150 mM NaCl, 1% Triton X-100, 25 mM β-glycerophosphate, 1 mM EDTA, 1 mM EGTA, 1 mM Na$_3$VO$_4$, and Complete Protease Inhibitor Cocktail. Immunoprecipitated proteins were eluted in 2X Laemmli sample buffer (containing 4% β-mercaptoethanol) by boiling for 5–10 min.

## Soft agar assays

$4 \times 10^3$ SUM159 or MCF7 cells were trypsinized, counted, and resuspended in 0.366% Bacto agar in complete medium and seeded on a layer of solidified 0.5% Bacto agar in 6-well plates 48 hr post-transfection. After solidification, another layer of complete medium containing 2X FBS was added. Cells were grown for 14 days (fed every 3 days) and then stained with 0.05% crystal violet in phosphate-buffered saline (PBS) overnight. Images were taken using a LEICA MZ75 stereomicroscope at a magnification of 6.3X and colony number was quantified using ImageJ software.

## Tumor expression and patient survival analysis

Normal/tumor expression data for ZNF446 and TRIM27 was generated by the TCGA Research Network (https://www.cancer.gov/tcga) and visualized with Oncomine and UALCAN, respectively (*Rhodes et al., 2004*; *Chandrashekar et al., 2017*). Survival analysis was performed using Kaplan Meier plotter (*Györffy et al., 2010*). For ZNF165, expression data was bifurcated using the upper quartile. For RRAD expression data was bifurcated using median and lower tertile for OS and DMFS, respectively. For ZNF446, expression was based on median values. Hazard ratios and p-values were calculated using the logrank test.

## Human breast tissue immunohistochemistry (IHC)

Normal breast and TNBC tissues were obtained with informed consent from the UTSW Tissue Management Shared Resource (TMSR) in compliance with the UTSW Internal Review Board committee. TRIM27 IHC was optimized and performed by the TMSR according to internal protocols using the Dako Autostainer Link 48 system. Quantification of nuclear TRIM27 signal was performed using IHC Profiler (ImageJ) according to the developer's guidelines (*Varghese et al., 2014*).

## Orthotopic xenografts

SUM159T-Luciferase cells ($5 \times 10^5$) stably expressing TRIM27 shRNAs or a non-targeting hairpin were injected in the mammary fat pad of 10 week old female NOD.Cg-*Prkdc$^{scid}$ Il2rg$^{tm1Wjl}$*/SzJ (NSG) mice obtained from the Jackson Laboratory as previously described (*Westcott et al., 2015*). Ten mice per condition were tested. Eight mice allowed for the comparison of the mean values of two different samples (Control and Test) with a standard deviation value of up to 60% of the difference between the means and a power of 0.8 using a two−sample t test. Ten mice were used to account for possible technical problems in handling samples after the mouse is sacrificed, the need to prematurely sacrifice mice because of an adverse response to surgical or non−surgical procedures, illness, or complete failure of tumor growth. Mice were assigned to groups randomly. Tumors

were measured weekly using a digital caliper and volume (V) was calculated using the following formula: $V = (Length \times (Width^2))/2$. Starting four weeks post injection, tumors were measured weekly with bioluminescent imaging (BLI). At six weeks post injection, mice were sacrificed after obtaining final BLI measurements. These studies were conducted in accordance with a UT Southwestern Institutional Animal Care and Use Committee (IACUC) approved protocol.

## Statistical analysis

Graphpad Prism (Graphpad Software) was used to perform statistical analyses. Data were assessed by one or two-tailed, unpaired t-tests or Mann-Whitney, or hypergeometric distribution. P-values less than 0.05 were considered significant. Biological replicates were considered to be experiments performed on distinct samples, while technical replicates were considered identical samples run in parallel.

## Acknowledgements

We thank the Center for Cancer Systems Biology (CCSB) at the Dana-Farber Cancer Institute for providing access to the human protein interactome database, and Melanie Cobb and Michael Kalwat for comments and suggestions related to this study. We would also like to acknowledge the assistance of the UTSW Tissue Management Shared Resource and the UTSW Small Animal Imaging Resource, which are supported in part by the Harold C Simmons Cancer Center through an NCI Cancer Center Support Grant (5P30CA142543). This work was supported by a donation from the families of Todd and Dawn Aaron and Steven L and Carol Aaron.

## Additional information

### Funding

| Funder | Grant reference number | Author |
|---|---|---|
| National Cancer Institute | 5R01CA196905 | Zane A Gibbs<br>Luis C Reza<br>Chun-Chun Cheng<br>Angelique W Whitehurst |
| National Institute of General Medical Sciences | 5T32GM007062 | Zane A Gibbs |

The funders had no role in study design, data collection and interpretation, or the decision to submit the work for publication.

### Author contributions

Zane A Gibbs, Conceptualization, Resources, Data curation, Formal analysis, Validation, Investigation, Visualization, Methodology, Writing - original draft, Writing - review and editing; Luis C Reza, Formal analysis, Investigation; Chun-Chun Cheng, Jill M Westcott, Kathleen McGlynn, Formal analysis, Investigation, Methodology; Angelique W Whitehurst, Conceptualization, Resources, Data curation, Supervision, Funding acquisition, Investigation, Visualization, Methodology, Writing - original draft, Writing - review and editing

### Author ORCIDs

Zane A Gibbs (iD) https://orcid.org/0000-0003-0294-1878
Angelique W Whitehurst (iD) https://orcid.org/0000-0002-9505-0240

### Ethics

Animal experimentation: This study was performed in strict accordance with the recommendations in the Guide for the Care and Use of Laboratory Animals of the National Institutes of Health. All of the animals were handled according to approved institutional animal care and use committee (IACUC) protocols (2016-101795) of UT Southwestern. All surgery was performed under anesthesia, and every effort was made to minimize suffering.

## Decision letter and Author response

Decision letter https://doi.org/10.7554/eLife.57679.sa1
Author response https://doi.org/10.7554/eLife.57679.sa2

---

# Additional files

## Supplementary files

• Supplementary file 1. GREAT (ver. 3.0.0) analysis of the 118 ZNF165/SMAD3 co-bound regions determined using default association rules.

• Supplementary file 2. Genes included in the EPITHELIAL_MESENCHYMAL_TRANSITION gene set with corresponding enrichment values for WHIM12 cells depleted of ZNF165.

• Supplementary file 3. Interactome data for ZNF165 and ZNF446 obtained from the CCSB Interactome Database.

• Supplementary file 4. Sequence-based reagents used for gene expression analysis, ChIP, and knockdown experiments.

• Transparent reporting form

## Data availability

Data have been submitted under GEO access code GSE130364.

The following dataset was generated:

| Author(s) | Year | Dataset title | Dataset URL | Database and Identifier |
|---|---|---|---|---|
| Gibbs Z, Whitehurst AW | 2020 | Genomic binding profiles for ZNF165, ZNF446, and SMAD3 in triple-negative breast cancer | http://www.ncbi.nlm.nih.gov/geo/query/acc.cgi?acc=GSE130364 | NCBI Gene Expression Omnibus, GSE130364 |

The following previously published datasets were used:

| Author(s) | Year | Dataset title | Dataset URL | Database and Identifier |
|---|---|---|---|---|
| Whitehurst A, Maxfield K | 2015 | ChIP-Seq analysis to identify direct binding of ZNF165 | https://www.ncbi.nlm.nih.gov/geo/query/acc.cgi?acc=GSE65937 | NCBI Gene Expression Omnibus, GSE65937 |
| Tufegdzic VA, Rueda OM, Vervoort SJ, Batra SA, Goldgraben MA, Uribe-Lewis S, Greenwood W, Coffer PJ, Bruna A, Caldas C | 2015 | Context-Specific Effects of TGF-$\beta$/SMAD3 in Cancer Are Modulated by the Epigenome. | https://www.ebi.ac.uk/ega/studies/EGAS00001001570 | EBI, EGAS00001001570 |
| Whitehurst A, Maxfield K | 2015 | Integrative Functional Characterization of Cancer-Testis Antigens Defines Obligate Participation in Multiple Hallmarks of Cancer | https://www.ncbi.nlm.nih.gov/geo/query/acc.cgi | NCBI Gene Expression Omnibus, GSE63986 |

---

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
