## [Decision Letter]

**Acceptance summary:**

This manuscript describes how the DNA-binding protein ZNF165, which is normally expressed in testis, is reactivated and co-opted in triple-negative breast cancer (TNBC) to promote tumor growth. Employing molecular studies and phenotypic assays (both in cell lines and mouse xenografts) the authors identified a TGFβ-SMAD3/4-ZNF165-ZNF446-TRIM27 axis through which a subset of genes become dysregulated to promote cancer progression. These results provide a foothold towards targeting normally-intractable TNBC through this novel regulatory complex.

**Decision letter after peer review:**

[Editors’ note: the authors submitted for reconsideration following the decision after peer review. What follows is the decision letter after the first round of review.]

Thank you for submitting your work entitled "The testis protein ZNF165 is a SMAD3 cofactor that coordinates oncogenic TGFβ signaling in triple-negative breast cancer" for consideration by *eLife*. Your article has been reviewed by four peer reviewers, one of whom is a member of our Board of Reviewing Editors, and the evaluation has been overseen by a Senior Editor. The reviewers have opted to remain anonymous.

Reviewers discussed the strengths and shortcomings of the current manuscript. While they appreciated the overall good quality of the work and the interesting area of research (i.e. role of testes-specific proteins in TGF β signaling and cancer), in the end they agreed not to invite a revised manuscript at this stage. Reviewers agreed that the first part of the paper did not provide enough new knowledge over a previous publication by the same team (Maxfield et al., 2015), and that the second part of the paper, which describes novel findings, was of lesser quality and more superficial (see appended reviews for specific concerns). After discussion, reviewers agreed that the experiments required to elevate the second part of the paper would require significantly more time than the usual two month period allowed at *eLife*. Therefore, they decided to recommend a Rejection of the current manuscript, while encouraging the authors to employ the constructive criticism to improve the manuscript, and potentially consider a new submission to *eLife* in the future.

Reviewer #1:

The manuscript by Gibs et al. demonstrates a novel role for the testes-specific DNA-binding protein ZNF165 in control of a subset of TGFβ/SMAD target genes in TNBC. The data demonstrates co-occupancy and co-regulation of some SMAD3 targets in TNBC cells, including genes required for anchorage-independent growth (e.g. RRAD). Mechanistically, the paper also describes other cofactors required for control of a subset of SMAD targets, including ZNF446 and TRIM27. Overall, the manuscript is (mostly) well written with well supported conclusions, but it relies heavily on in vitro cell-based models with no support for this novel mechanism in mouse models of TNBC or available data from human tumors.

Are ZNF165 and its binding partners ZNF446 and TRIM27 required for growth in xenograft or PDX models? What is the clinical behavior of TNBCs that express high or low levels of these three factors? (e.g. prognosis, 5 year survival). What is the relationship between expression of these three factors and TGFβ signatures in TNBC? In sum, the manuscript falls short of making a compelling case for the mechanism described to be relevant in the clinical setting, something that could be addressed with additional experiments and analyses of existing datasets.

Also, the passage of the manuscript about characterization of ZNF446 and TRIM27 is a bit confusing, showing data for interaction with SMAD3 and SMAD4, respectively. Does ZNF446 interact with SMAD4? Does TRIM27 interact with SMAD3? The authors seem to have all reagents to complete the proximity ligation assays and co-immunoprecipitation to answer these questions.

Reviewer #2:

Gibbs et al. demonstrate here that the Cancer-Testes antigen ZNF165 is a co-factor for SMAD3 in regulation of TGF-β signaling in TNBC. Evidence based on genomic binding profiles is compelling. The linking to gene expression analyses of these genomic binding data identify a set of genes that have both ZNF165 and SMAD3 binding sites AND are regulated by TGF-β. So far, so good. The authors then select a few of these target genes for functional analyses. This is a less compelling part of the manuscript. For instance, the authors show that knockdown of RRAD impairs soft agar growth of SUM159 cells, but fail to address whether 2D growth of these cells is equally affected, as appears the case in WHIM12 cells. A quick survey of the Broad Institute project Achilles shows that knockdown of RRAD is not particularly essential in breast cancer compared to a number of other cancers (https://depmap.org/portal/gene/RRAD?tab=dependency). Yet the authors argue that their functional data provide evidence that "RRAD is a relevant transcriptional target of TGFβ in TNBC, as evidenced by its essentiality for anchorage independent growth".

This statement implies that in all cells in which RRAD is an essential gene, its expression is driven by TGF-β and in the absence of TGF-β, knockdown of RRAD has little effect (due to low expression). These conclusions are in my view not supported by the data presented.

I am fully on board with the notion that ZNF165 is a SMAD3 co-factor and involved in TGF-β mediated transcriptional responses. The presence of ZNF446 and TRIM27 in these complexes is also well-supported. In my view, the authors should refrain from over-interpreting these data. There is no compelling evidence in this manuscript that these cancer-testis antigens are key components of the switch from tumor suppressive to oncogenic signalling by TGF-β.

Reviewer #3:

In general, I find that the topic of this paper is inherently of high interest. CT antigens are known to occur in tumors, but mechanistic details have not been worked out for most of them. Add to that the fact that this paper is trying to address this gap in triple-negative breast cancer cells-which elude common treatments-the findings from such a study will have major consequences on the cancer field. However, I find the paper slightly shaky in two areas: the importance of Figure 2C to the overall conclusions, and the frequent use of tagged proteins and overexpression to draw conclusions.

Specific concerns:

1) It would be interesting to know what percentage of SMAD3 binding sites are promoters, and compare that with the analogous percentage from ZNF165:SMAD3 co-bound sites. In other words, are co-bound sites more/less enriched in terms of promoter regions vs all SMAD3 binding sites? The authors already have the datasets, and so will just have to look.

2) Do SMAD3 and ZNF165 have binding sites on each other's genetic regions? This is assumed not to be the case, but should be explicitly addressed because it directly affects the interpretation of scatter plots like Figure 2B. This is also the case for other similar plots i.e. Figure 4C and Figure 5C.

3) I find Figure 2C quite problematic in terms of interpretation, and definitely not as straightforward as the authors put it in the text. In terms of the raw magnitude of relative expression (i.e. y-axis of scatter plot), the siZNF165 condition is behaving as you would expect from Figure 2B. However, the trend increase/decrease in expression is sometimes not what you would expect from Figure 2B: e.g. SMURF2, according to Figure 2B, is repressed both by SMAD3 and ZNF165, but its expression increases with increased exposure to TGFβ (and presumably increased SMAD3 signaling), making this panel inconsistent with Figure 2B.

4) Most of the co-immunoprecipitation experiments and the proximity ligation assays were done with tagged constructs and overexpression. It is well-known that a major caveat of overexpression is promiscuous binding at non-physiological concentrations. The authors already show in Figure 5E that they are able to carry out a proximity ligation assay with antibodies towards endogenous proteins. Why not do the same for the other PLA experiments and co-IP experiments?

5) In the PLA experiments, control experiments with the TGFβ inhibitor (SB-431542) should always be done. This is only done in Figure 3.

6) In Figure 3E, technically, what is shown is that there is proximal interactions between two proteins. But whether this is mediated by DNA binding (which is the assumption here) is not explicitly tested. It would be good if a PLA experiment can be done with a ZNF165 version that does not bind DNA, to show that these interactions are due to ZNF165's ability to bind DNA binding sites, and not because ZNF165 can bind to SMAD3 by itself. This is also the case for other PLA experiments in the study.

Reviewer #4:

The work of Gibbs et al. dissects a transcriptional regulator complex, which is formed in triple negative breast cancer (TNBC) cell lines. The authors have previously identified a major player in this regulatory process (ZNF165) and are now describing that:

1) Binding sites of ZNF165 and SMAD3 largely overlap and found a common GC-rich element

2) ZNF165 acts upstream of TGF-β sensitive genes

3) SCAND1 and ZNF446 form a complex with ZNF165

The work is largely focussed on dissecting the individual interaction partners and network regulation and touches upon effects in cell growth.

My major concerns are that:

1) much of the novelty has been already published in the previous study of the group (Maxfield, 2015)

2) signalling cascades or proposed working model should highlight what is known and what is the new contribution (especially with regards to TGF-β), how this relates to early and late breast cancer settings and how the observations fit the testis-specific regulatory program (perhaps a model summarizing the work can be added to the last figure).

Other concerns:

1) Add survival analysis of the factors mentioned (TNBC vs. non- TNBC, early and late stage using TGCA data) that will support the speculative roles in early and late cancer responses

2) Motif enrichment (Figure 1G): similar motifs has been identified (PB0164.1 for SMAD3). I therefore question whether the motif is a genuine ZNF165 binding motif. It appears rather common for many SMAD3 binding sites.

3) How does the TGF-β specific response influence the readout? Why were 16h chosen (Figure 2A) when much of the responses happen already within the first 2h (Figure 2C)?

[Editors’ note: further revisions were suggested prior to acceptance, as described below.]

Thank you for submitting your article "The testis protein ZNF165 is a SMAD3 cofactor that coordinates oncogenic TGFβ signaling in triple-negative breast cancer" for consideration by *eLife*. Your article has been reviewed by three peer reviewers, one of whom is a member of our Board of Reviewing Editors, and the evaluation has been overseen by Kevin Struhl as the Senior Editor. The reviewers have opted to remain anonymous.

The reviewers have discussed the reviews with one another and the Reviewing Editor has drafted this decision to help you prepare a revised submission.

Summary:

The work of Gibbs et al. describes how the DNA-binding protein ZNF165, which is normally expressed in testis, is “re-activated” in triple-negative breast cancer (TNBC) to promote tumor growth. Employing molecular studies and phenotypic assays (both in cell lines and mouse xenografts) the authors identified a TGFβ-SMAD3/4-ZNF165-ZNF446-TRIM27 axis through which a subset of genes become dysregulated to promote cancer progression.

The current, revised version of the manuscript is much improved relative to the original submission. Key improvements and additions include new evidence of physical interaction between the various combinations of testis-specific transcription factors and their cofactors, new ChIP-seq datasets reinforcing the interplay between these factors on chromatin, additional analysis of human tumor gene expression data, and TNBC xenograft data showing a requirement for TRIM27 for cancer cell proliferation. The authors have made a commendable work at addressing the major and minor concerns raised by reviewers of the original manuscript.

Overall, the reviewers agreed that the manuscript merits publication in *eLife* after addressing the points below. The topic of the paper was deemed of high interest in general, because through this work, the authors provide a foothold towards targeting normally-intractable TNBC through the testis-specific DNA-binding protein ZNF165. Additional value was found in the interesting biology described, whereby ZNF165, when expressed ectopically, can nucleate the assembly of a new transcription regulatory complex in cancer cells.

Revisions:

1) Whenever possible, please create VENN diagrams that are proportional in size to comparisons made.

2) Similar to Figure 1—figure supplement 1B, please add a VENN diagram for ZNF165 (at least for WHIM12 and SUM159 cells for which data are presented in this manuscript)

3) Please explain why those 118 ZNF165 and SMAD3 co-bound regions (Results paragraph one) were inspected further. Next to which genes do they reside? Are these the 65 TGFβ-responsive genes mentioned in Results paragraph two and in Figure 2—figure supplement 1A? If so, please refer to them in the text. If not, please add a supplementary table with those gene names.

4) Figure 1F: do the co-occupied sites correspond to the 118 sites mentioned above? What is the total number of SMAD3 binding sites? Please add an "n=XX" in the figure legend.

5) Please provide a supplementary table containing the genes used in the gene set enrichment analysis (Figure 1—figure supplement 1D)

6) Add a label (roman letter) for each of the quadrants as discussed in the manuscript (Figure 2A, Figure 6D)

7) Mention previously that the role of SMURF2 was previously addressed (Maxfield et al., 2015; Ohashi et al., 2005) either in paragraph two of subsection “ZNF165 and SMAD3 cooperate to modulate TGFβ-responsive gene expression” or in the Discussion.

8) Subsection “ZNF165 physically associates with SMAD3 in a TGFβ-dependent manner”: Comment on that ZNF165 does not regulate the SMAD3 and SMAD4 gene loci.

9) Move Figure 5C/D to supplements. It is good to show that the peaks are overlapping in part but the actual genes (Figure 5I) are more relevant.

10) Materials and methods:

a) Ideally add lot number of the antibodies used for the ChIP experiments.

b) Add machine name for the IP (also a Bioruptor?)

11) The main story is about how the nucleating of a new complex that is capable of tuning transcriptional activity, but most of the measurements for such "transcriptional" activity is assayed by steady state mRNA levels, e.g. qRT-PCR of target genes, after siRNA knockdown of factors of interest. In the case of Figure 2A, this is still believable because the data in panel A is at least supported by the H3K27ac ChIP in panels E and F. However, in other parts of the paper, e.g. Figure 5J, changes in expression levels (as measured by qRT-PCR) is taken to be due to transcriptional changes-which is not a given- as any post-transcriptional effects that are caused by the siRNA knockdowns would also lead to changes in steady state expression levels. The same point applies to Figure 6E. It is important then for the authors to introduce a clear disclaimer stating that post-transcriptional effects cannot be discarded, and that a fraction of the genes altered by the siRNA knockdowns could be non-transcriptional targets.

12) Figure 1—figure supplement panel E: The labelling of the two lines appear to be flipped. According to the text, patients with high ZNF165 expression are associated with greater metastatic potential, but this is inconsistent with the graph shown in panel E, as currently labelled.

13) Figure 2A: Quadrants are not labelled, but are referred to in the text.

14) "anchorage-independent growth using soft agar assays"… The figure panel referred to here should just be Figure 6—figure supplement panel E only. Panel D in the supplement refers to something else, and it is unclear what panel D is trying to show.

---

## [Author Response]

[Editors’ note: the authors resubmitted a revised version of the paper for consideration. What follows is the authors’ response to the first round of review.]

Reviewers discussed the strengths and shortcomings of the current manuscript. While they appreciated the overall good quality of the work and the interesting area of research (i.e. role of testes-specific proteins in TGF β signaling and cancer), in the end they agreed not to invite a revised manuscript at this stage. Reviewers agreed that the first part of the paper did not provide enough new knowledge over a previous publication by the same team (Maxfield et al., 2015), and that the second part of the paper, which describes novel findings, was of lesser quality and more superficial (see appended reviews for specific concerns). After discussion, reviewers agreed that the experiments required to elevate the second part of the paper would require significantly more time than the usual two month period allowed at eLife. Therefore, they decided to recommend a Rejection of the current manuscript, while encouraging the authors to employ the constructive criticism to improve the manuscript, and potentially consider a new submission to eLife in the future.

The key concern of the original manuscript was a lack of robust and in-depth analysis of the novel findings. We believe that the revised manuscript contains a more comprehensive and detailed study responding to the following points brought up by the reviewers:

1) Multiple reviewers requested evidence of endogenous interactions between members of the transcriptional complex. We now present proximity ligation assays demonstrating endogenous associations between ZNF165 and SMAD3, ZNF165 and SMAD4, ZNF446 and SMAD3, ZNF446 and SMAD4, TRIM27 and SMAD3, and TRIM27 and SMAD4. We also present endogenous immunoprecipitations of ZNF165 and ZNF446. Together, we think that these assays present a compelling case that these proteins form a complex within intact cells, a novel observation that has important implications for understanding TGFβ signaling in TNBC.

2) To further reinforce the functional relevance of the ZNF165- SMAD3-ZNF446 interaction, we generated ChIP-seq datasets for ZNF446. This dataset demonstrates co-occupancy of all three of these factors on chromatin and reinforces the notion that ZNF165 and SMAD3 form a complex with ZNF446 to modulate TGFβ-induced transcription. Notably, a prior attempt from another group to define ZNF446 binding sites indicated that either ZNF446 lacked widespread DNA binding activity or was unable to bind in that context2. Thus, the dataset generated herein provides the first indepth analysis of ZNF446 genomic occupancy and, along with the associated protocols, should provide essential information for future functional analysis of ZNF446 in different contexts.

3) The initial manuscript had limited information regarding the clinical relevance of the transcriptional complex in human tumors. To mitigate this concern, we now present multiple lines of evidence that these factors are associated with poor outcome. First, we demonstrate that ZNF165 correlates with a reduced distant metastasis free survival in breast cancer. In addition, we find that ZNF165 expression is significantly associated with an EMT gene expression signature, a hallmark of tumor cells that exhibit oncogenic TGFβ signaling. Second, we show that ZNF446 is elevated in breast cancer and trends with reduced overall survival. Third, we find that TRIM27 is also elevated in TNBC and that its nuclear localization is associated with tumorigenic state. Fourth, expression of RRAD, a target gene of the ZNF165 transcriptional complex, exhibits a dramatic association with poor outcome in mesenchymal breast cancers (enriched for TNBC). However, we observed no association with patient outcome in ER+ breast tumors, where TGFβ signaling is tumor-suppressive. Finally, to test the relevance of the transcriptional complex in vivo, we depleted TRIM27 from TNBC cells and found a significant reduction in tumor growth at the orthotopic site in mice.

4) We have also addressed a number of the minor concerns of the reviewers including the removal of Panel 2C, additional statistical analysis of the ChIP-seq data sets, and reduction in overinterpretations regarding a role for this complex in switching the TGFβ signaling output from tumor-suppressive to oncogenic.

[Editors’ note: what follows is the authors’ response to the second round of review.]

Overall, the reviewers agreed that the manuscript merits publication in eLife after addressing the points below. The topic of the paper was deemed of high interest in general, because through this work, the authors provide a foothold towards targeting normally-intractable TNBC through the testis-specific DNA-binding protein ZNF165. Additional value was found in the interesting biology described, whereby ZNF165, when expressed ectopically, can nucleate the assembly of a new transcription regulatory complex in cancer cells.Revisions:1) Whenever possible, please create VENN diagrams that are proportional in size to comparisons made.

Size adjustments have been made to the VENN diagrams for the following: Figure 1—figure supplement 1 (panels A, B, C, and E), Figure 5 (panels C, D, and I), Figure 5—figure supplement 1 (panels B, C, and D).

2) Similar to Figure 1—figure supplement 1B, please add a VENN diagram for ZNF165 (at least for WHIM12 and SUM159 cells for which data are presented in this manuscript)

A VENN diagram displaying the overlap between ZNF165 ChIP-seq peaks identified in WHIM12 and SUM159 cells has been added to the Figure 1 supplement (Figure 1—figure supplement 1A). In addition, VENN diagrams showing the overlap between ZNF165 and SMAD3 ChIP-seq peaks in both WHIM12 and SUM159 cells have also been added here (Figure 1—figure supplement 1B and 1C) for additional clarity.

3) Please explain why those 118 ZNF165 and SMAD3 co-bound regions (Results paragraph one) were inspected further. Next to which genes do they reside? Are these the 65 TGFβ-responsive genes mentioned in Results paragraph two and in Figure 2—figure supplement 1A? If so, please refer to them in the text. If not, please add a supplementary table with those gene names.

The 118 co-bound regions are those found in WHIM12 cells bound by both ZNF165 and SMAD3 as identified by ChIP-seq peak overlap. This point is now made more clearly in the text (Results paragraph two) with the inclusion of a Venn diagram (Figure 1—figure supplement 1B) and a supplementary table (Supplementary file 1) as requested. These 118 co-bound sites were used solely to identify any motifs unique to co-bound regions in the WHIM12 cells. These are not the 65 TGFβ-responsive genes that we used to identify co-regulated genes.

4) Figure 1F: do the co-occupied sites correspond to the 118 sites mentioned above? What is the total number of SMAD3 binding sites? Please add an "n=XX" in the figure legend.

The co-occupied sites in Figure 1F correspond to the 118 overlapping peaks described in Figure 1F. For full clarification on this and point 3, we now refer to these as “ZNF165-SMAD3 co-bound sites” in Figure 1F and in the text. As requested, the legend for figure 1F has now been updated to read: “Pie charts displaying the distribution of genomic features bound by ZNF165-SMAD3 (n=118) or only SMAD3 (n=27,979) in WHIM12 cells.”

5) Please provide a supplementary table containing the genes used in the gene set enrichment analysis (Figure 1—figure supplement 1D)

Supplementary file 2 has been created with the genes used in the gene set enrichment analysis, along with their corresponding enrichment values used to generate Figure 1—figure supplement 1G. The table is referenced within the figure legend for Figure 1—figure supplement 1G.

6) Add a label (roman letter) for each of the quadrants as discussed in the manuscript (Figure 2A, Figure 6D)

Labels for the quadrants in Figure 2A and Figure 6D have been added. In addition, the text was modified to call out the appropriate quadrants mentioned for Figure 6D (subsection “TRIM27 is essential for ZNF165 transcriptional activity and tumor growth in vivo”).

7) Mention previously that the role of SMURF2 was previously addressed (Maxfield et al., 2015; Ohashi et al., 2005) either in paragraph two of subsection “ZNF165 and SMAD3 cooperate to modulate TGFβ-responsive gene expression” or in the Discussion.

We now mention that the role of SMURF2 was previously addressed and that, together with the data presented in this study, ZNF165 and SMAD3 cooperate to relieve negative feedback within the TGFβ pathway via repression of SMURF2.

8) Subsection “ZNF165 physically associates with SMAD3 in a TGFβ-dependent manner”: Comment on that ZNF165 does not regulate the SMAD3 and SMAD4 gene loci.

We have added a comment at the end of the section describing Figure 3 that ZNF165 does not regulate the SMAD3 or SMAD4 gene loci. This is based on inspection of the ChIP-seq datasets presented in this study and our previous work, as referenced.

9) Move Figure 5C/D to supplements. It is good to show that the peaks are overlapping in part but the actual genes (Figure 5I) are more relevant.

Figure 5C and 5D have been moved to the supplement (Figure 5—figure supplement 1B and 1C). In addition, VENN diagrams of target gene overlaps between ZNF165 and ZNF446 have been added to Figure 5 (panels C and D). The main text has been updated to reflect these changes.

10) Materials and methods:a) Ideally add lot number of the antibodies used for the ChIP experiments.b) Add machine name for the IP (also a Bioruptor?)

Lot numbers for antibodies used in ChIP experiments have been added to the Materials and methods section. Additionally, the name of the sonicator used for the endogenous IP experiment has been added to the Materials and methods section.

11) The main story is about how the nucleating of a new complex that is capable of tuning transcriptional activity, but most of the measurements for such "transcriptional" activity is assayed by steady state mRNA levels, e.g. qRT-PCR of target genes, after siRNA knockdown of factors of interest. In the case of Figure 2A, this is still believable because the data in panel A is at least supported by the H3K27ac ChIP in panels E and F. However, in other parts of the paper, e.g. Figure 5J, changes in expression levels (as measured by qRT-PCR) is taken to be due to transcriptional changes-which is not a given- as any post-transcriptional effects that are caused by the siRNA knockdowns would also lead to changes in steady state expression levels. The same point applies to Figure 6E. It is important then for the authors to introduce a clear disclaimer stating that post-transcriptional effects cannot be discarded, and that a fraction of the genes altered by the siRNA knockdowns could be non-transcriptional targets.

We thank the reviewers for bringing this important concern to our attention and we fully agree. As requested, we have added a sentence within the Discussion section pointing out this caveat.

12) Figure 1—figure supplement panel E: The labelling of the two lines appear to be flipped. According to the text, patients with high ZNF165 expression are associated with greater metastatic potential, but this is inconsistent with the graph shown in panel E, as currently labelled.

The point of this plot was to demonstrate the early drop-off in distant metastasis free survival (up to 150 months) in the ZNF165-high patients. At later time points, these lines do appear flipped but only represent relatively few remaining patients. Importantly, the ZNF165- high population having a greater metastatic potential is best reflected in the hazardratio of 1.4 which is calculated on the entire data set (p=0.004).

13) Figure 2A: Quadrants are not labelled, but are referred to in the text.

Quadrants have been added to the figure panels (see above).

14) "anchorage-independent growth using soft agar assays"… The figure panel referred to here should just be Figure 6—figure supplement panel E only. Panel D in the supplement refers to something else, and it is unclear what panel D is trying to show.

The purpose of Figure 6—figure supplement 1D was to demonstrate that there is a mild defect observed in 2-dimensional growth of TNBC cell lines following TRIM27 loss, according to DepMap. In contrast, when cells are placed in the more stressful soft-agar environment, we observed a dramatic growth defect. To avoid confusion, we have removed panel D from the figure supplement as these data do not add any appreciable information to the manuscript. We appreciate the reviewers bringing this point to our attention. The text has been updated to reflect this change.